# Detecting Backdoor Samples in Contrastive Language Image Pretraining

**Hanxun Huang[1]   Sarah Erfani[1]   Yige Li[2†]  Xingjun Ma[3†]  James Bailey[1]**

[1]School of Computing and Information Systems, The University of Melbourne, Australia
[2]School of Computing and Information Systems, Singapore Management University, Singapore
[3]School of Computer Science, Fudan University, China
{hanxun,sarah.erfani,baileyj}@unimelb.edu.au;
{yigeli}@smu.edu.sg;{xingjunma}@fudan.edu.cn.

## Abstract

Contrastive language-image pretraining (CLIP) has been found to be vulnerable to poisoning backdoor attacks where the adversary can achieve an almost perfect attack success rate on CLIP models by poisoning only 0.01% of the training dataset. This raises security concerns on the current practice of pretraining large-scale models on unscrutinized web data using CLIP. In this work, we analyze the representations of backdoor-poisoned samples learned by CLIP models and find that they exhibit unique characteristics in their local subspace, i.e., their local neighborhoods are far more sparse than that of clean samples. Based on this finding, we conduct a systematic study on detecting CLIP backdoor attacks and show that these attacks can be easily and efficiently detected by traditional density ratio-based local outlier detectors, whereas existing backdoor sample detection methods fail. Our experiments also reveal that an unintentional backdoor already exists in the original CC3M dataset and has been trained into a popular open-source model released by OpenCLIP. Based on our detector, one can clean up a million-scale web dataset (e.g., CC3M) efficiently within 15 minutes using 4 Nvidia A100 GPUs. The code is publicly available in our GitHub repository.

## 1 Introduction

Contrastive language-image pretraining (CLIP) (Radford et al., 2021) is a popular self-supervised learning framework (Chopra et al., 2005; Hadsell et al., 2006; Grill et al., 2020; Chen & He, 2021; Caron et al., 2021; Bardes et al., 2022) that allows pretraining of large-scale multi-modal models on web data without human annotations (Radford et al., 2021; Jia et al., 2021). However, in a recent study by Carlini & Terzis (2022), it was found that CLIP is extremely vulnerable to poisoning backdoor attacks, where an attacker backdoors the victim model by poisoning (adding the trigger to) a few training samples (Gu et al., 2017; Chen et al., 2017; Liu et al., 2018). Carlini & Terzis (2022) investigated backdoor attack on CLIP with a patch trigger, and revealed that an attacker can successfully attack CLIP by poisoning only 0.01% of the training samples. This poisoning rate is marginal compared to supervised learning where successful attacks generally require a high poisoning rate of 1% – 10%. Poisoning existing web-scale datasets is also realistic since the curator typically only maintains a list of hyperlinks to the image. Carlini et al. (2024) have shown that adversaries could poison 0.01% of web-scale datasets by purchasing expiring domains with $10 USD. This vulnerability poses a major security threat to the current practice of CLIP, considering that many popular multi-modal models (Alayrac et al., 2022; Liu et al., 2023; Betker et al., 2023; Awadalla et al., 2023) were pre-trained using CLIP on unscrutinized web data crawled from untrusted sources.

Several backdoor defence techniques for CLIP have been proposed, which are mostly robust training methods using heavy data augmentations (Bansal et al., 2023; Yang et al., 2023a), or a uni-modal objective (Yang et al., 2024). These methods train the model directly on the poisoned training dataset while minimizing the effect caused by the backdoor samples. Although these methods have demonstrated promising results, an in-depth understanding of the unique characteristics of CLIP

---

[†]Corresponding author.

backdoor attacks is absent in the current literature. A more concerning fact is that no backdoor sample detection method exists that can help data owners and model developers efficiently clean up a million-scale web dataset. A backdoor sample detection method is essential for secure CLIP because (1) it can detect and remove backdoor-poisoned samples from a large-scale dataset once and for all, and (2) it can help remove noisy or unintentional backdoor samples from the dataset even when there are no attacks. Moreover, it has been shown theoretically that the detection and removal of backdoor data is equivalent to robust training under mild assumptions (Manoj & Blum, 2021). However, prior detection works against supervised backdoor attacks revealed that detecting backdoor samples under extremely low poisoning rates (e.g., 0.01%) is very difficult (Hayase et al., 2021; Huang et al., 2023). Moreover, existing backdoor sample detection methods (Chen et al., 2018; Tran et al., 2018; Gao et al., 2019; Hayase et al., 2021; Li et al., 2021; Hou et al., 2024) were all developed for supervised learning, which might not be applicable for CLIP.

In this paper, we explore the local neighborhood characteristics of backdoor samples in the representation space and discover one major weakness of CLIP backdoor attacks, i.e., they have a much more sparse neighborhood than clean samples, making them outliers. Based on this finding, we further investigate the detectability of CLIP backdoor samples by both existing backdoor sample detection methods and traditional outlier detection methods. Surprisingly, we find that traditional general-purpose outlier detection methods can detect CLIP backdoor samples with high accuracy, while existing backdoor sample detection methods for supervised learning could fail in certain cases. In particular, classic methods such as distance to the $k$-th nearest neighbor and isolation forest (iForest) (Liu et al., 2008) can outperform existing backdoor sample detection methods (Li et al., 2021; Huang et al., 2023). Performance can be further improved by considering density-focused techniques, such as the simplified local outlier factor (SLOF) (Schubert et al., 2014) and dimensionality-aware outlier detection (DAO) (Anderberg et al., 2024).

Our main contributions are as follows:

- We present a systematic study on the detectability of poisoning backdoor attacks on CLIP, and show that existing detection methods designed for supervised learning can fail on CLIP.
- We reveal one major weakness of CLIP backdoor samples related to the sparsity of their representation local neighborhood, which facilitates highly accurate and efficient detection using efficient local density-based detectors. With these detectors, one can clean up a million-scale poisoned dataset (e.g., CC3M) within 15 minutes using 4 Nvidia-A100 GPUs.
- Our experiments in the clean setting reveal that there exist unintentional (natural) backdoors in the CC3M dataset, which has been injected into a popular open-source model released by OpenCLIP.

## 2  RELATED WORK

**Backdoor Attacks.** The objective of a backdoor attack is to deceive a victim model to learn a shortcut correlation between the trigger and a targeted output. The adversary can subsequently manipulate the predictions of the victim model at the test time with the trigger. Based on the attackers' and defenders' capabilities, existing backdoor attacks can be categorized into *data-poisoning* and *training-manipulation* attacks. In data-poisoning attacks, the adversary injects triggers into the defender's training data, but the defender has full control of the model training. Such attacks simulate the scenario where the defender utilizes an untrusted web dataset for training. In training manipulation attacks (Lin et al., 2020; Shumailov et al., 2021; Bagdasaryan & Shmatikov, 2021; Nguyen & Tran, 2021; Doan et al., 2021; Wang et al., 2022), the attacker can manipulate both the training data and the objective function to implant the trigger and then releases the backdoored model for the victim to download. This simulates the scenario where the victim downloads pre-trained model parameters from untrusted open-source platforms. The focus of this work is data poisoning attacks.

Existing backdoor attacks are mostly focused on attacking supervised learning (Gu et al., 2017; Chen et al., 2017; Liu et al., 2018). For poisoning attacks, the trigger pattern is one main contributing factor to the success of the attack. The trigger pattern could be a patch (Gu et al., 2017) or a blending image (Chen et al., 2017). Advanced attacks leverage more complex trigger patterns such as periodical patterns (Barni et al., 2019), natural reflections (Liu et al., 2020), physical objects (Li et al., 2020; Wenger et al., 2021), adversarial perturbations (Turner et al., 2018; Zhao et al., 2020),

GANs (Cheng et al., 2020), Instagram filters (Liu et al., 2019), image generator (Sun et al., 2024) and image frequency perturbations (Zeng et al., 2021; Li et al., 2023). While injecting the trigger pattern into training images, the attacker could either alter the corresponding label (known as *dirty-label* attacks) or keep the label unchanged (known as *clean-label* attacks) (Turner et al., 2018; Zhao et al., 2020). There could also be multiple triggers released by one or more adversaries for the same dataset (Li et al., 2024b), which is a realistic setting for downloading data from untrusted sources.

Carlini & Terzis (2022) proposed the first poisoning backdoor attacks on CLIP with patch triggers. Compared to supervised learning, poisoning backdoor attacks on CLIP can achieve a high attack success rate at a much lower poisoning rate (i.e., 0.01%). Concurrently, targeted poisoning attack in the finetuning stage (Yang et al., 2023b) and training-manipulation backdoor attacks have also been developed for CLIP (Jia et al., 2022; Liu et al., 2022; Tao et al., 2023). The main focus of our work is detecting poisoning backdoor samples in CLIP, for which we follow the same threat model and experimental setup as Carlini & Terzis (2022).

**Backdoor Defense.** Backdoor defense can be categorized into 1) trigger synthesis, 2) backdoor model detection, 3) robust training, and 4) backdoor sample detection methods. Trigger synthesis aims to recover the trigger patterns used to poison and attack the victim model (Liu et al., 2019; Wang et al., 2019; Hu et al., 2022). Model detection aims to determine if a trained model contains backdoors (Chen et al., 2019; Kolouri et al., 2020; Xu et al., 2021; Feng et al., 2023; Kuang et al., 2024). Note that trigger synthesis and model detection methods will still need backdoor removal techniques to obtain a robust model. A robust training strategy aims to (pre)train a backdoor-free model on backdoor-poisoned dataset by robustifying the training procedure of supervised learning (Li et al., 2021; Borgnia et al., 2021; Huang et al., 2022; Dolatabadi et al., 2022), self-supervised learning (Li et al., 2024a) or CLIP (Bansal et al., 2023; Yang et al., 2024; 2023a).

Backdoor sample detection determines if a data point is infected with the backdoor trigger. It can leverage either the statistics of the deep features (Tran et al., 2018; Chen et al., 2018; Tang et al., 2021), sensitivity characteristics to certain perturbations and transformations (Gao et al., 2019; Chen et al., 2022; Hou et al., 2024) or inference time detection with contrastive prompting (Niu et al., 2024). Cognitive Distillation (CD) extracts a minimal pattern that allows the model to produce the same output and uses the norm of the extracted mask to detect whether a training/test sample is backdoored (Huang et al., 2023). However, it is an optimization-based method that is time-consuming and of limited scalability for web-scale datasets. Anti-Backdoor Learning (ABL) tracks sample-specific training loss during training and detects samples of the lowest loss as backdoor samples (Li et al., 2021). The above defense methods were all developed under supervised learning, with many relying on the class labels to function, which is not available in CLIP. SafeCLIP is proposed as end-to-end robust training strategy to obtain a backdoor-free model from potentially poisoned dataset (Yang et al., 2024). It has two components: one for detecting backdoor data and one for robust training on safe and risky subsets.

**Outlier Detection.** Outlier detection is a classic problem in data mining. It aims to find data points that deviate from the general distribution. It can be categorized into parametric and non-parametric approaches. The parametric approach makes explicit assumptions about the nature of the underlying data distribution (Yang et al., 2009; Satman, 2013), while the non-parametric does not. The non-parametric approach is more suitable for unsupervised settings such as backdoor sample detection. It include statistical methods (Goldstein & Dengel, 2012; Li et al., 2022b), and ensemble methods (Lazarevic & Kumar, 2005; Zhao et al., 2021) such as the isolation forest (iForest) (Liu et al., 2008). Local outlier methods are another type of non-parametric outlier detection methods, such as k-nearest-neighbor (Ramaswamy et al., 2000), local outlier factor (LOF) (Breunig et al., 2000), and their improved versions (Tang et al., 2002; Papadimitriou et al., 2003; Latecki et al., 2007; Kriegel et al., 2008). These methods either explicitly or implicitly assess the density in the vicinity of a query point (Campos et al., 2016) and data points with low density are usually regarded as outliers. Local intrinsic dimensionality (LID) is another local measure that describes the growth rate of the number of data points in the vicinity of the query point (Levina & Bickel, 2004; Houle, 2017). LID has been used in various machine learning-related applications (Gong et al., 2019; Ansuini et al., 2019; Pope et al., 2021). Notably, it is used in outlier detections (Houle et al., 2018), detecting adversarial examples (Ma et al., 2018a) and backdoor samples (in a supervised setting) (Dolatabadi et al., 2022). In this work, based on our empirical observation, we choose to exploit simplified local outlier factor (SLOF) (Schubert et al., 2014) and its extension dimensionality-aware outlier detection (DAO) (Anderberg et al., 2024) for detecting backdoor samples.

## 3 PRELIMINARIES

In this section, we first describe poisoning backdoor attacks on CLIP and then introduce three outlier detection metrics explored in this work, including simplified local outlier factor (SLOF), local intrinsic dimensionality (LID), and dimensionality-aware outlier detection (DAO).

### 3.1 POISONING BACKDOOR ATTACKS ON CLIP

Following existing work (Carlini & Terzis, 2022), we focus on multi-modal Contrastive Language Image Pretraining (CLIP) (Radford et al., 2021), which learns a joint representation of images and text from image-text data. Given a image-text dataset $\mathcal{D} \subset \mathcal{X} \times \mathcal{T}$ that contains pairs of $(\boldsymbol{x}_i, \boldsymbol{t}_i)$, where $\boldsymbol{x}_i$ is an image, and $\boldsymbol{t}_i$ is the associated descriptive caption. The CLIP framework uses an image encoder $f_I : \mathcal{X} \mapsto \mathbb{R}^d$ and a text encoder $f_T : \mathcal{T} \mapsto \mathbb{R}^d$, and projects the image and text to a joint representation space $\mathbb{R}^d$. The image representation can be obtained by $\boldsymbol{z}_i^x = f_I(x_i)$ and the text representation is $\boldsymbol{z}_i^t = f_T(t_i)$. For a given batch of $N$ image-text pairs $\{\boldsymbol{x}_i, \boldsymbol{t}_i\}_{i=1}^N$, CLIP adopts the following training loss function:

$$-\frac{1}{2N}\sum_{j=1}^N \log \frac{\exp(\text{sim}(\boldsymbol{z}_j^x, \boldsymbol{z}_j^t)/\tau)}{\sum_{k=1}^N \exp(\text{sim}(\boldsymbol{z}_j^x, \boldsymbol{z}_k^t)/\tau)} - \frac{1}{2N}\sum_{k=1}^N \log \frac{\exp(\text{sim}(\boldsymbol{z}_k^x, \boldsymbol{z}_k^t)/\tau)}{\sum_{j=1}^N \exp(\text{sim}(\boldsymbol{z}_j^x, \boldsymbol{z}_k^t)/\tau)},$$

where $\tau$ is a trainable temperature parameter, and $\text{sim}(\cdot)$ is a similarity measure. The first term in the above objective function contrasts the images with the texts, while the second term contrasts the texts with the images.

The main focus of our work is detecting poisoning backdoor images in the CLIP pretraining dataset, as most existing backdoor triggers have been concentrated in the vision domain (Carlini & Terzis, 2022). Poisoning images is also generally more practical than text in web-scale pretraining datasets (Carlini et al., 2024). We adopt the same threat model and setup as Carlini & Terzis (2022). In addition to backdoor attacks, we also aim to detect poisoned data in targeted data poisoning attacks (Biggio et al., 2012; Carlini & Terzis, 2022; Yang et al., 2023b).

**Backdoor Attack (BA).** For poisoning backdoor attack on CLIP, the adversary could use a function $A(\cdot)$ to construct a backdoored image-text pair $(\boldsymbol{x}', \boldsymbol{t}') = A((\boldsymbol{x}, \boldsymbol{t}))$. The trigger pattern can be inserted into the image using $\boldsymbol{x}' = \boldsymbol{m} \odot \boldsymbol{\Delta} + (1 - \boldsymbol{m}) \odot \boldsymbol{x}$, where $\odot$ is the element-wise multiplication and $\boldsymbol{\Delta}$ is a trigger pattern. This is a general definition of backdoored image commonly adopted in existing work (Wang et al., 2019). For the associated caption $\boldsymbol{t}' \in$ caption set, one might use engineered prompt templates (Radford et al., 2021) as the caption set, such as "*a photo of a {target}*", where *target* is the attacker's desired output. The attacker could also insert the trigger to the image where the *target* is already in the captions $\boldsymbol{t} = \boldsymbol{t}'$ (without text caption modification), which is known as a clean-label attack on CLIP. We assume the adversary can inject the poisoned subset $\mathcal{D}_b = \{(\boldsymbol{x}_i', \boldsymbol{t}_i') = A(\boldsymbol{x}_i, \boldsymbol{t}_i) | \boldsymbol{t}_i' \in \text{caption set}\}_{i=1}^M$ into defender's training data. The attacker's objective is to control the model to produce a desired output. For example, in the case of using engineered prompt templates for zero-shot classification, the attack is successful if the adversary queries the victim model with a backdoor image $\boldsymbol{x}'$ and receives its desired backdoor *target* as prediction.

**Targeted Data Poisoning Attack (TDPA).** Unlike backdoor attacks, targeted data poisoning attacks aim to fool the model by misclassifying a specific sample $\boldsymbol{x}'$ into a targeted class $y_t$ without using a universal trigger $\Delta$. The poisoned subset is $\mathcal{D}_b = \{(\boldsymbol{x}', \boldsymbol{t}_i') | \boldsymbol{t}' \in \text{caption set}\}_{i=1}^M$ and the caption set is constructed by finding all captions in $\mathcal{D}$ that contains target keyword $y_t$. The adversary's goal is to misclassify $\boldsymbol{x}'$ into $y_t$ in the zero-shot classification. Similar to backdoor attacks, we assume the adversary can poison the training data.

### 3.2 OUTLIER DETECTION METRICS

We will empirically show in Section 4 that the local neighborhood of a backdoor-poisoned sample is much more sparse (low density) than that of clean samples. This motivates us to exploit local and density-based outlier detection metrics to differentiate CLIP backdoor samples. While other metrics could also be worth investigating, we focus on the following three classic metrics.

**Simplified Local Outlier Factor (SLOF).** The SLOF (Schubert et al., 2014) is a variant of the classical Local Outlier Factor (LOF) (Breunig et al., 2000). The 'local outlier' refers to a query

point $\boldsymbol{q}$ that is sufficiently different from other neighboring points in its vicinity. LOF considers the typical density ratio. For the query point $\boldsymbol{q}$, if it is less dense than its neighborhoods, then it is more likely to be an outlier. The classical LOF is based on the reachability distances that require multiple levels of neighborhood computation. The SLOF provides a simplified version by using the distance to the $k$-th nearest neighbor, defined as the following:

$$\text{SLOF}_k(\boldsymbol{q}) \triangleq \frac{1}{k} \sum_{\boldsymbol{o} \in \text{NN}_k(\boldsymbol{q})} \frac{k\text{-dist}(\boldsymbol{q})}{k\text{-dist}(\boldsymbol{o})},$$

where $k\text{-dist}(x)$ is the distance to a sample $x$'s $k$-th nearest neighbor and $\text{NN}_k(\cdot)$ are the $k$ nearest neighbors.

**Local Intrinsic Dimensionality (LID).** The LID metric (Houle, 2017) describes the rate of growth in the number of data objects encountered as the distance from the reference sample increases. It measures the intrinsic dimensionality in the vicinity of the query point. Formally, let $F$ be a real-valued function that is non-zero over some open interval containing $r \in, r \neq 0$.

**Theorem 1 (Houle (2017))** *If $F$ is continuously differentiable at $r$, then*

$$\text{LID}_F(r) \triangleq \frac{r \cdot F'(r)}{F(r)}.$$

We are interested in functions $F$ that satisfy the conditions of a cumulative distribution function (CDF). The LID at a query point is defined as the limit when the radius r tends to zero:

$$\text{LID}_F^* \triangleq \lim_{r \to 0^+} \text{LID}_F(r).$$

Henceforth, when we refer to the LID of a function $F$, or of a point $\mathbf{x}$ whose induced distance distribution has $F$ as its CDF, we will take 'LID' to mean the quantity $\text{LID}_F^*$. In practice, the LID needs to be estimated, such as using maximum likelihood estimation (MLE) (Levina & Bickel, 2004) or Bayesian estimation (Joukhadar et al., 2024). We refer to the estimated value as $\widehat{\text{LID}^*}$.

**Dimensionality-Aware Outlier Detection (DAO).** DAO (Anderberg et al., 2024) is a criterion that extends LOF and SLOF using theory in dimensionality characteristics. Specifically, DAO is defined as the following:

$$\text{DAO}_k(\boldsymbol{q}) \triangleq \frac{1}{k} \sum_{\boldsymbol{o} \in \text{NN}_k(\boldsymbol{q})} \left( \frac{k\text{-dist}(\boldsymbol{q})}{k\text{-dist}(\boldsymbol{o})} \right)^{\widehat{\text{LID}^*_{F_o}}}.$$

A DAO score greater than 1 indicates it is likely to be an outlier. It suggests that the query point $\boldsymbol{q}$ has a local probability measure too small to be consistent with those of its neighbors in the domain. The DAO criterion is a generalization of SLOF. In essence, SLOF implicitly assumes the underlying local intrinsic dimensionalities are equal to 1 ($\text{LID}^*_{F_o} = 1$) for all data points. This may not always be realistic for machine learning applications (Ma et al., 2018a;b; Gong et al., 2019; Ansuini et al., 2019; Pope et al., 2021; Huang et al., 2024; Zhou et al., 2024). As a result, DAO is theoretically more favorable than SLOF.

## 4 DETECTING CLIP BACKDOOR ATTACKS

In this section, we begin by discussing the problem definition of backdoor sample detection. We then show an intuitive example of backdoor representations as local outliers. Finally, we present the exploration of SLOF, LID, and DAO for CLIP backdoor sample detection.

### 4.1 BACKDOOR SAMPLE DETECTION

**Threat Model.** Following previous works (Carlini & Terzis, 2022), we assume the attacker can poison the defender's training data but does not have access to the training process. The defender has full control over the training process but has no prior knowledge of (i) the poisoning rate, (ii) the trigger pattern, (iii) the target, or (iv) whether an image-text pair is clean or backdoored. The defender aims to produce the probability of an image-text pair being poisoned.

**Problem Formulation.** We denote the training data as $\mathcal{D} = \mathcal{D}_c \cup \mathcal{D}_b$, the clean subset as $\{(\boldsymbol{x}_i, \boldsymbol{t}_i)\}_{i=1}^N \in \mathcal{D}_c$, and the poisoned subset as $\{(\boldsymbol{x}_i', \boldsymbol{t}_i')\}_{i=1}^M \in \mathcal{D}_b$, respectively. The poisoning rate is defined as $\frac{|\mathcal{D}_b|}{|\mathcal{D}|} = \frac{M}{M+N}$. The defender's goal is to accurately detect pairs $(\boldsymbol{x}, \boldsymbol{t}) \in \mathcal{D}_b$.

Backdoor sample detection is a binary classification task ('backdoor' vs 'clean'). We consider the following function $g(\cdot)$ to determine whether an image-text pair $(\boldsymbol{x}_i, \boldsymbol{t}_i)$ contains backdoor based on the detection score $s_i$:

$$g(\boldsymbol{x}_i, \boldsymbol{t}_i) = \begin{cases} 1 & \text{if } s_i > t, \\ 0 & \text{if } s_i \leq t, \end{cases} \tag{1}$$

where, $t$ is a threshold, $g(\cdot) = 1$ indicates a backdoor sample and $g(\cdot) = 0$ indicates a clean sample. In practice, the defender can adjust $t$ based on the statistics of the detection score, such as the mean and standard deviation. Alternatively, the defender might remove certain percentages of data from the training set. For accurate detection, the most crucial objective is to correctly rank the score $s$ within the dataset, e.g., assign higher scores to backdoor samples.

## 4.2 CHARACTERIZING CLIP BACKDOOR SAMPLES

Our goal is to find unique characteristics of CLIP backdoor samples. We find existing methods, such as ABL (Li et al., 2021) and CD (Huang et al., 2023), are not sufficient to find the distinctive differences between clean and backdoor samples in CLIP. In this work, we take a different approach to examine the learned representations of the model. To motivate our approach, in Figure 1a, we provide an illustrative example of a representation with CLIP trained on backdoor poisoned data using the patch trigger (Gu et al., 2017). CLIP uses the contrastive learning loss that clusters image-text pairs with similar contents to the same region. All the backdoor-poisoned samples contain similar features (the trigger) and are likely to be clustered together in a particular region. Since the trigger is a strong signal and the model is overconfident about these poisoned samples, the surrounding subspace has distinctive characteristics compared to clean samples, e.g., they are tightly clustered and far away from other clean data. This can be observed in Figure 1a.

As a result, to detect poisoned backdoor samples, one might consider using local distance measures, such as the distance to the $k$-th nearest neighbors, the $k$-dist. Consider randomly sampling a batch of the data (batch size 1024), if the poisoning rate is 0.01% and there is 1 poisoned sample in the batch, the probability of the rest of the data being clean is $0.9999^{1023}$. The $k$-th nearest neighbor is highly likely to be a clean sample, a larger $k$-dist. For the clean data, it is likely the clean sample as well, results in a smaller $k$-dist. This characteristic makes backdoor representations as outliers. In Figure 1b, we show a controlled experiment. As the poisoning rate increases, the distribution of the $k$-dist stays fairly stable for the clean samples but dramatically decreases for the backdoor samples. This indicates that as long as the poisoning rate is low, the $k$-th nearest neighbor for the backdoor representation is a clean representation, and the backdoor representation is an outlier in the batch. Hence, with an appropriate locality $k$, simple $k$-dist is sufficient to detect these poisoned samples.

An alternative criterion is to use local density measures, where locality is given by $k$ nearest neighbors, whose distance is used to estimate the density. As shown in Figure 1a, data points within the backdoor region are less dense compared to clean data points, which can be measured with the density ratio. The 'local outlier' (backdoor data point in this case) is sufficiently different from observations in its vicinity. In classical outlier detection literature, this can be characterized by the LOF metric (Breunig et al., 2000), SLOF (Schubert et al., 2014) and DAO (Anderberg et al., 2024). Using Figure 1a as an example, local density outlier detection considers the radius of the $k$-neighborhood ball of the query data point (red and green circle with solid line) over the radius of the $k$-neighborhood ball of each its nearest neighbors (circles with dash line). A higher ratio indicates the point of interest is less dense than its neighbors and thus likely to be an outlier.

## 4.3 DETECTING CLIP BACKDOOR SAMPLES

In this subsection, we describe how to apply local outlier detection methods to detect CLIP backdoor samples. At a high level, we train a model using an untrusted dataset $\mathcal{D}$, then iterate over each training data point to extract their representations. We randomly sample a batch of data from the dataset and then apply outlier detection methods. For each data point, it will produce a score $s$ to indicate its probability of being a backdoor sample. We present the pseudo-code in Appendix A.

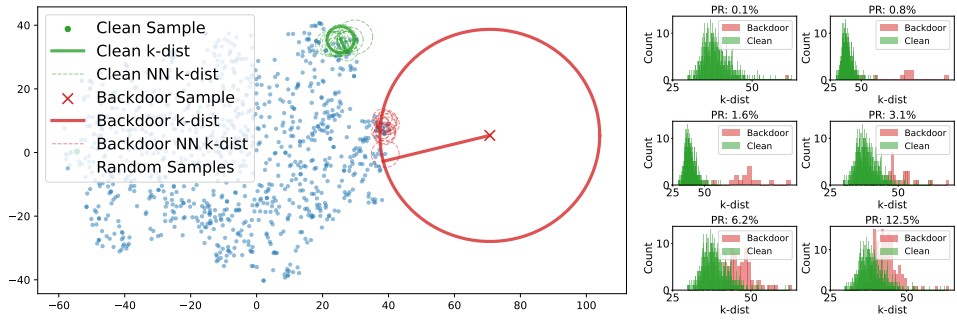

(a) A t-SNE visualization of the learned representations by CLIP    (b) Distribution of $k$-dist

Figure 1: (a) The CLIP learned presentations are projected into a 2-D space using t-SNE. The red cross is a backdoor data point, the green dot is a clean data point, and the blue dot is a randomly sampled data point. The $k$-dist is the distance to the $k$-th nearest neighbor, and the circle with the solid line is the region containing all $k$ nearest neighbors. The circle with a dashed line is the region containing $k$ nearest neighbors for the $k$-th neighbors. $k$ is set to 16. (b) The distribution $k$-dist for clean and backdoor data with different poisoning rates (PR) within a batch.

Once the backdoor score is obtained for each data point, one might simply remove data that has abnormally high scores or remove a certain percentage of the data point according to the score. A practitioner could use the remaining safe subset to train the model from scratch with standard training to obtain a backdoor-free model or alternatively use a robust training strategy as a double security measure. The most important aspect is to accurately detect backdoor samples, which is the main focus of this work. Since removing too much data from the training set might degrade the performance of the model, the score for backdoor samples should be distinctively higher compared to the clean data. Otherwise, if we remove a small portion of the untrustworthy data points, it might not be sufficient to remove all poisoned samples.

## 5    EXPERIMENTS

For all our experiments, we adopt the open-source implementation of CLIP (i.e., OpenCLIP) (Il-harco et al., 2021), ResNet-50 (RN50) (He et al., 2016), and ViT-B-16 (Dosovitskiy et al., 2021) as the vision encoder, and choose hyperparameters following existing works (Carlini & Terzis, 2022). Details are in Appendix B.1. We conduct our experiment on the CC3M dataset (Sharma et al., 2018). The evaluation is conducted on ImageNet (Deng et al., 2009) with the zero-shot classifier (Radford et al., 2021). Note that evaluation with CC3M and ImageNet is recommended by Carlini & Terzis (2022) for studying the backdoor poisoning attacks against CLIP. We provide evaluations with a larger dataset, the CC12M (Changpinyo et al., 2021), in Appendix B.7, which shows a consistent performance for local outlier methods.

For the CLIP single trigger backdoor attack (STBA), we follow the existing work by Carlini & Terzis (2022) which uses a $16 \times 16$ patch trigger (Gu et al., 2017) and 'banana' as the target label. We also explore commonly used triggers in supervised learning, including Blend with a hello kitty image(Chen et al., 2017), periodical signal pattern (SIG) (Barni et al., 2019), image filter with Nashville style (Liu et al., 2019), WaNet (Nguyen & Tran, 2021), and BLTO (Sun et al., 2024). Additionally, we evaluate multiple trigger setting (Li et al., 2024b), where attacker(s) can release multiple triggers (MTBA) to attack a single target or multiple targets. We use 3 triggers: the Patch, Nashville style, and WaNet. The multi-trigger settings are denoted as MT-S (single target) and MT-M (multiple targets). We also investigate the clean-label setting (Turner et al., 2018), where the patch trigger is only inserted into images with a caption already containing the target. For targeted data poisoning attack (TDPA), we randomly select an image and construct the caption set with captions in the training set that contains the keyword 'banana'. In terms of poisoning rate, existing work (Carlini & Terzis, 2022) already demonstrates that a 0.01% poisoning rate is sufficient for the patch trigger and TDPA on the ResNet-50. For other triggers, we conducted a coarse grid search to find the minimal poisoning rate to guarantee a high attack success rate (ASR).

For backdoor sample detection, we evaluate local outlier detection ($k$-dist, LID, SLOF, DAO), comparing it with other backdoor data detection methods and the classical global outlier detection method isolation forest (iForest) (Liu et al., 2008). Note that since most of the existing detection methods are based on supervised learning, we only include the state-of-the-art methods that are applicable to CLIP, including ABL (Li et al., 2021) and Cognitive Distillation (CD) (Huang et al., 2023). For LID outlier detection (Houle et al., 2018), we use the maximum likelihood estimation as estimator (Levina & Bickel, 2004). We also compare with the backdoor data detection components in SafeCLIP (Yang et al., 2024), and follow their hyperparameter setting. On the same hardware setting, it would take 15 minutes for local outlier methods to run the detection, while CD costs 11.2 hours and ABL takes 4.1 hours. See Appendix B.1 for more details on the experimental settings. We use the area under the ROC curve (AUC) as the main performance metric. The AUC can be seen as the probability a backdoor sample has a higher score than a normal sample. We provide the analysis on the sensitivity to $k$ for all local outlier methods in Appendix B.2. It shows that $k$-dist, SLOF, and DAO are robust to different values of $k$. Additional results using the false positive rate at 95% true positive rate (FPR@95) in Appendix B.7, demonstrating findings consistent with those presented in this section. Yang et al. (2024) discussed that SafeCLIP is not robust to poisoning rates higher than 0.5%. In Appendix B.5, we demonstrate that local outlier detection methods can consistently identify poisoned data even with a poisoning rate of up to 10%.

## 5.1 Detection Performance Evaluation

Table 1 compares the detection performance of different outlier detection methods and shows among all, the local outlier detection methods, $k$-dist, SLOF, and DAO are the most effective and efficient, even compared to dedicated backdoor sample detection methods ABL, CD and SafeCLIP. The iForest also achieves a non-trivial performance. The LID detection, however, is not robust to different triggers and architectures. Overall, the results indicate that CLIP backdoor samples are indeed evident outliers in the representation space and can be effectively detected.

Table 1: Comparing the AUC (%) of different outlier detection methods against different backdoor attacks. The poisoning rates are minimized based on a coarse grid search to guarantee a non-trivial attack success rate (ASR). Clean Acc (CA) and ASR are measured by the top-1 zero-shot accuracy (%) on ImageNet. The best results are **boldfaced**.

| Vision Encoder | Threat Model | Trigger | Poisoning Rate (%) | CA (%) | ASR (%) | ABL | CD | Safe CLIP | LID | iForest | $k$-dist | SLOF | DAO |
|---|---|---|---|---|---|---|---|---|---|---|---|---|---|
| RN50 | STBA | Patch | 0.01 | 17.0 | 100.0 | 27.86 | 97.17 | 83.42 | 99.29 | 99.73 | 99.75 | **99.86** | **99.86** |
| | | Clean Label | 0.07 | 17.1 | 95.0 | 63.50 | 48.68 | 46.93 | 88.23 | 94.01 | 96.75 | **97.10** | 97.06 |
| | | Nashville | 0.1 | 16.7 | 78.7 | 61.69 | 98.33 | 46.37 | 61.07 | 99.35 | 99.51 | 99.61 | **99.62** |
| | | WaNet | 0.1 | 16.2 | 83.8 | 56.07 | 99.19 | 85.82 | 57.07 | 99.55 | 99.82 | **99.85** | **99.85** |
| | | Blend | 0.1 | 16.8 | 75.9 | 60.07 | 99.64 | 57.01 | 54.94 | 99.80 | 99.82 | **99.88** | **99.88** |
| | | SIG | 0.1 | 16.3 | 67.3 | 56.88 | 99.06 | 82.15 | 54.03 | 99.62 | 99.67 | **99.69** | **99.69** |
| | | BLTO | 0.1 | 16.7 | 98.3 | 58.88 | 97.60 | 84.25 | 43.04 | 99.81 | 99.85 | **99.86** | **99.86** |
| | MTBA | MT-S | 0.1 | 16.5 | 79.5 | 47.32 | 99.34 | 80.98 | 94.87 | 99.59 | 99.59 | 99.66 | **99.67** |
| | | MT-M | 0.1 | 16.2 | 74.7 | 53.24 | 95.33 | 78.10 | 97.11 | 98.32 | 98.50 | 98.74 | **98.76** |
| | TDPA | - | 0.01 | 16.8 | 100.0 | 55.37 | **99.99** | 81.71 | 80.60 | 99.95 | 99.96 | 99.96 | 99.96 |
| ViT B-16 | STBA | Patch | 0.1 | 15.2 | 99.8 | 29.19 | 8.40 | 85.88 | 45.36 | 88.39 | **98.48** | 96.42 | 95.27 |
| | | Clean Label | 0.07 | 15.7 | 19.0 | 57.78 | 50.30 | 44.24 | 70.28 | 54.73 | 69.22 | **71.48** | 70.82 |
| | | Nashville | 0.1 | 15.7 | 41.4 | 58.91 | 95.36 | 61.77 | 25.52 | 92.41 | **99.06** | 97.71 | 96.83 |
| | | WaNet | 0.1 | 15.2 | 12.2 | 34.53 | 45.69 | 84.60 | 38.65 | 89.81 | **98.45** | 96.59 | 95.59 |
| | | Blend | 0.1 | 15.7 | 95.8 | 65.43 | 96.36 | 47.57 | 12.92 | 89.24 | **99.68** | 88.43 | 84.50 |
| | | SIG | 0.1 | 15.3 | 82.9 | 57.20 | 80.05 | 61.35 | 17.60 | 89.82 | **99.33** | 97.33 | 96.06 |
| | | BLTO | 0.1 | 6.4 | 13.2 | 29.19 | 77.87 | 83.39 | 88.84 | 78.26 | 91.24 | **94.36** | 94.34 |
| | MTBA | MT-S | 0.1 | 15.3 | 28.5 | 23.85 | 61.89 | 82.05 | 73.16 | 87.66 | 95.50 | 96.59 | **96.61** |
| | | MT-M | 0.1 | 15.2 | 36.2 | 30.29 | 51.70 | 79.64 | 77.92 | 73.11 | 78.28 | 86.16 | **86.81** |
| | TDPA | - | 0.01 | 15.5 | 100.0 | 58.79 | 54.63 | 92.03 | 79.00 | 81.88 | 95.71 | **98.51** | 98.16 |

Compared to SafeCLIP, local outlier methods are more robust to different trigger types. SafeCLIP can achieve 83% to 85% on some triggers (Patch, WaNet, and SIG), local outlier methods can consistently reach 97% to 99% for various attacks. Interestingly, compared to ViT-B-16, RN50 shows slightly better detection performance for local outlier methods. In practice, since the defender controls the training process, they can use RN50 for detection to remove potentially poisoned data from the dataset, after which the purified dataset can be used to train any other encoders.

Comparing the two dedicated backdoor sample detection methods for supervised learning, ABL and CD, the latter one shows a considerably better performance. However, it performs badly against clean-label attacks using ResNet-50, with only 48.68% AUC. A possible explanation is that in a dirty-label setting, all the captions are changed according to the template, and the model learns a strong signal (keyword) that behaves similarly to supervised learning. In a clean-label setting, however, the captions are unchanged, and the model learns a diverse set of captions. This difference might affect the optimization of the mask for CD. ABL uses sample-specific loss values for the detection score, e.g., a lower loss value indicating a backdoor. However, in every iteration of contrastive learning, the data points are sampled randomly. Thus, there could be noise in the loss as the distance to negative pairs depends on the randomly sampled data.

## 5.2 Outlier Filtering as a Defense

It has been theoretically shown that robust learning on an untrustworthy dataset is equivalent to the effective detection and removal of poisoned data points (Manoj & Blum, 2021). Given accurate detection, this simple strategy can effectively mitigate the threat of backdoor attack. However, determining how many samples to remove is a challenging question. Here, we experiment to remove 10% of the data from a poisoned CC3M dataset according to the detection score using DAO. To test the defense effect of filtering, we retrain the CLIP model from scratch on the remaining 90% of the purified data. We plot the distributions of the DAO scores of backdoor vs. clean samples in Figure 2. The 'Threshold' line marks the 10% cutoff point, where samples on the left side of the line will be kept while those on the right will be removed from the dataset. The score distributions of other detection metrics are provided in Appendix B.7. The performance and robustness of the retrained model are reported in Table 2.

Table 2: Defense performance of backdoor sample filtering using DAO for filtering rate 10% on poisoned CC3M. The results are presented in the form of clean zero-shot accuracy (%) / attack success rate (%) on the ImageNet validation set. Results are based on the ResNet-50 as the vision encoder.

| Dataset | Patch | Clean Label | Nashville | WaNet | Blend | SIG | MT-S | MT-M | TDPA |
|---|---|---|---|---|---|---|---|---|---|
| Poisoned | 17.0 / 100.0 | 17.1 / 95.0 | 16.7 / 78.7 | 16.2 / 83.8 | 16.8 / 75.9 | 16.3 / 67.3 | 16.5 / 79.5 | 16.2 / 74.7 | 16.8 / 100.0 |
| Purified | 16.2 / 0.0 | 16.7 / 0.0 | 16.1 / 9.6 | 16.7 / 0.5 | 15.8 / 0.6 | 16.4 / 0.2 | 16.7 / 0.6 | 16.4 / 15.00 | 16.3 / 0.0 |

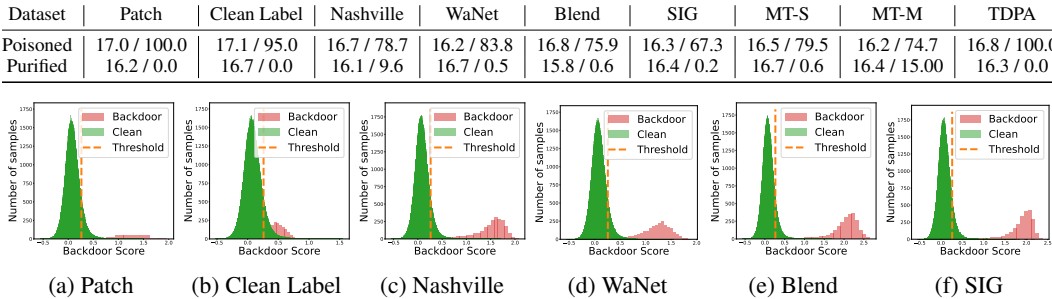

(a) Patch  (b) Clean Label  (c) Nashville  (d) WaNet  (e) Blend  (f) SIG

Figure 2: The distributions of the DAO detection score on poisoned CC3M using ResNet-50 as the vision encoder.

As shown in Table 2, removing outlier data points can significantly reduce the ASR to less than 1%, except for Nashville and MT-M, which are slightly higher. This defense effectiveness is somewhat expected, as it shows in Figure 2 that most backdoor samples are separable from the clean ones. Interestingly, filtering 10% of the data points does not significantly affect the clean performance. Additional results for zero-shot classification and linear probing on other datasets are in Appendix B.3. They are consistent with the results shown in this section. This indicates the existence of a considerable proportion of noisy data and anomalies in web-crawled datasets like CC3M. Through the above experiment, we highlight the necessity and benefit of purifying a large-scale web dataset using local outlier detectors like SLOF and DAO. In Figure 3a, we present the sensitivity of the defense performance to different filtering percentages. For the patch trigger, removing 1% is sufficient to mitigate the backdoor threat. Additional results for other triggers are in Appendix B.3. They show that removing 5% is sufficient to mitigate the backdoor threat of different kinds of triggers. In practice, the defender can dynamically adjust the threshold to achieve a suitable performance-security trade-off. Alternatively, backdoor detection can be combined with a robust training strategy (Yang et al., 2023a; 2024) to train on the purified (safe) and removed (risky) subsets.

## 5.3 DETECTING UNINTENTIONAL BACKDOORS IN CC3M

With the local outlier detectors, we show that they can be applied to detect noisy samples and even unintentional backdoor attacks from the CC3M dataset (Sharma et al., 2018). We extract the representation of each CC3M image using our pre-trained CLIP and rank the outlierness of the images based on their DAO outlier score. We then manually check the top-ranked images and identify two types of anomalies: 1) meaningless images and 2) suspicious images with extremely high occurrences.

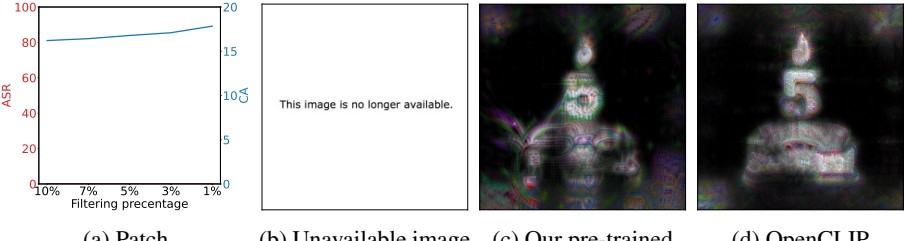

| (a) Patch | (b) Unavailable image | (c) Our pre-trained | (d) OpenCLIP |

Figure 3: (a) Defence performance for varying filtering percentages. (b) An example of the unavailable images. (c-d) The recovered trigger pattern of the birthday cake image on our pre-trained CLIP (b) and a model (c) released by OpenCLIP that uses ResNet-50 as the vision encoder.

One example of meaningless images is the *"this image is no longer available."* image. One example is illustrated in Figure 3b. Note that the captions and URLs of these images are still available, which inevitably causes a mismatch between the image content and the caption. This is the reason why removing outliers from CC3M does not affect the CLIP's performance (see Table 2). This indicates that even if the dataset is not poisoned, local outlier methods could still be applied to identify and remove the noisy samples.

We noticed several "suspicious" images that have similar content and the same caption *"the birthday cake with candles in the form of number icon."* We cannot show these examples here due to license restrictions. However, one can search online with the caption to see what these images look like. These images appear 798 times in the dataset, which roughly accounts for 0.03% of the entire dataset. We suspect this image is a natural (unintentional) backdoor trigger and has been learned into models trained on the Conceptual Captions dataset. To confirm this conjecture, we utilize two models: 1) our pre-trained CLIP model on CC3M, and 2) one pre-trained model released by OpenCLIP (Ilharco et al., 2021) which uses ResNet-50 as the vision encoder and is trained on CC12M (Changpinyo et al., 2021). We apply an adapted trigger recovery method based on Neural Cleanse (Wang et al., 2019) to distill a trigger pattern from both our pre-trained CLIP model and the OpenCLIP released model. The technical details of the trigger recovery method are described in Appendix B.4. We reveal the recovered triggers in Figure 3c and 3d, respectively. It shows that the birthday cake trigger has been successfully recovered on both models. The trigger recovered from our pre-trained CLIP and the OpenCLIP pre-trained model can achieve an ASR of 92.38% and 98.92% when attached to ImageNet test images in zero-shot classification. This not only confirms the existence of unintentional backdoors in web-scale datasets but also their possible existence in popular open-source multi-modal models.

## 6 CONCLUSION

In this work, we studied the local neighborhood characteristics of poisoning backdoor attacks on CLIP. We revealed one unique characteristic of CLIP backdoor attacks, which is related to the sparsity of their local representation subspace caused by the low poisoning rate. Based on this finding, we showed that traditional local outlier detection methods like SLOF and DAO can effectively detect different types of backdoor triggers. With the detectors, one can filter out poisoning backdoor data and noisy images from the CC3M dataset, and all can be done efficiently within 15 minutes using 4 Nvidia A100 GPUs and achieve near-perfect detection performance. Finally, we showed the existence of unintentional backdoor attacks in web-crawled datasets, which have already been pre-trained into popular open-source models. Our work verifies the necessity and benefit of data purification and we hope it may help inspire further research toward secure data curation and CLIP.

## REPRODUCIBILITY STATEMENT

There are two factors that impact the reproducibility of this work. The first factor is whether it is possible to reproduce the results. To facilitate this, we will make the source code openly available. However, due to the dynamic nature of web-scale datasets, some URLs may expire, making it challenging to reproduce the exact clean accuracy. Despite this, the attack success rate and detection results will remain unaffected. In this work, we successfully reproduced the results reported by Carlini & Terzis (2022), except for clean accuracy, as we could not access the complete CC3M dataset. We were only able to obtain 2.3 million image-text pairs from the CC3M dataset due to expired URLs. With the open-source code, evaluation results on the attack success rate and detection performance are fully reproducible.

The second factor is the computational resources required. Carlini & Terzis (2022) thoroughly investigated backdoor poisoning attacks against CLIP and recommended the best experimental setup, which we followed, using the CC3M dataset for evaluation in Section 5.1 and 5.2. However, pre-training still demands significant computational power, requiring approximately 100 GPU hours (on Nvidia A100) per attack. In our experiments, we expanded the evaluation beyond patch triggers used by Carlini & Terzis (2022) to include a broader range of backdoor triggers. This extension, covering 9 types of attacks with 2 types of encoders, requires an estimated 1,800 GPU hours. The detection evaluations including baseline methods took additional 288 GPU hours ($16 \times 18$), and filtering and retraining the models required 900 GPU hours. Results presented in Appendix B.7 for running detection on CC12M requires 400 GPU hours per attack, and detection requires 64 GPU hours. We evaluated 8 backdoor attacks, which would require 3,712 GPU hours. Therefore, to fully reproduce the results in this paper, we estimate a total of 6,700 GPU hours would be necessary.

## ACKNOWLEDGMENT

Xingjun Ma is in part supported by the National Key R&D Program of China (Grant No. 2021ZD0112804) and the National Natural Science Foundation of China (Grant No. 62276067). Sarah Erfani is in part supported by Australian Research Council (ARC) Discovery Early Career Researcher Award (DECRA) DE220100680. This research was supported by The University of Melbourne's Research Computing Services and the Petascale Campus Initiative.

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

## A    BACKDOOR SAMPLE DETECTION ALGORITHM

---

**Algorithm 1** Backdoor Sample Detection With Local Outlier Methods

---

1: **Input:** Vision encoder $f_I$, Text encoder $f_T$, Dataset $\mathcal{D}$, locality parameter $k$
2: $f = \text{train}(f_I, f_T, \mathcal{D})$              ▷ Standard training on an untrustworthy dataset.
3: **for** $i$ **to** $length(\mathcal{D})$ **do**
4:      $(\boldsymbol{x}, \boldsymbol{t}) = \text{sample}(\mathcal{D})$      ▷ Random sample a batch of data point from the dataset
5:      $\boldsymbol{z}^x = f_I(\boldsymbol{x})$                ▷ extract vision representations
6:      $\boldsymbol{z}^t = f_T(\boldsymbol{t})$                 ▷ extract text representations
7:      $\boldsymbol{z} = \text{concatenate}(\boldsymbol{z}^x, \boldsymbol{z}^t)$      ▷ Reference points for neighborhood selection
8:      $z_i = f(\boldsymbol{x_i})$           ▷ extract representations for the image of interest
9:      $s_i = \text{detection}(z_i, \boldsymbol{z}, k)$          ▷ $k$-dist, SLOF, LID, or DAO
10: **end for**
11: **Output:** backdoor score $s$

---

In Algorithm 1 (line No.9), we use the image embedding as the query point and all other image embeddings and text embeddings as reference points. The LID has been used in detecting adversarial examples (Ma et al., 2018a) as well as detection of backdoor data with class-wise reference points (Dolatabadi et al., 2022), which is not possible in CLIP. In Algorithm 1, instead of using class-wise reference points, we use random sample points for LID. In both works, it has been shown that a higher LID score for a query point means it is more likely to be a backdoor or adversarial example. Similarly, for SLOF and DAO, the higher the score, the more likely the data point is a backdoor sample. As a result, we directly use the LID or the outlier factor as the score to determine if the data point is poisoned.

For all outlier detection methods and LID, we use minibatch sampling to generate scores for each data point due to efficiency. While using the entire dataset is possible, it can be prohibitively costly for large-scale datasets. An alternative approach is to treat each batch of data as a sliding window in data streams and apply techniques like iLOF (Pokrajac et al., 2007) or MiLOF (Salehi et al., 2016). Existing literature suggests that minibatch sampling is sufficient to characterize the local neighborhood (Ma et al., 2018a). Our empirical evaluations also support this, indicating that using randomly sampled subsets as reference points is adequate for local outlier detection methods.

## B    EXPERIMENTS

In this section, we present our experimental setting in Appendix B.1, the sensitivity study to the locality parameters for local outlier methods in Appendix B.2, and additional filtering as a defense result in Appendix B.3. In Appendix B.4, we present the detailed descriptions for the trigger synthesis method used to obtain the results in Section 5.3. Finally, we show the sensitivity study to a higher poisoning rate in Appendix B.5 and additional results for the detection performance in Appendix B.7.

### B.1    EXPERIMENT SETTING

We conducted our experiments on Nvidia A100GPUs with PyTorch implementation. Each experiment distribution is conducted with data distributed in a parallel setting across 4 GPUs. We used automatic mixed precision due to its memory efficiency. Open-source code is available here[*].

For all experiments, we chose hyperparameters following existing work (Carlini & Terzis, 2022) and the open-source implementation OpenCLIP[†] (Ilharco et al., 2021). We use a learning rate of 0.001, with AdamW optimizer (Loshchilov & Hutter, 2019), weight decay is set to 0.2, batch size of 1024 and train for 30 epochs. We use ResNet-50 (He et al., 2016) and ViT (Dosovitskiy et al., 2021) for the image encoder and transformer (Vaswani et al., 2017) for the text encoder. The embedding dimension is set to 1024 for ResNet-50 and 512 for ViT. We use the same data augmentation as the implementation by OpenCLIP. We conduct our experiment on CC3M (Sharma et al., 2018) dataset.

---

[*]https://github.com/HanxunH/Detect-CLIP-Backdoor-Samples
[†]https://github.com/mlfoundations/open_clip

Due to the expired and invalid links, we only obtained 2.3M image-text pairs, so the reproduced clean performance is slightly lower than the result reported by OpenCLIP. This is normal as the number of data can significantly affect the performance of CLIP (Radford et al., 2021). The evaluation is conducted on ImageNet (Deng et al., 2009) with a zero-shot classifier using the prompt template (Radford et al., 2021).

For backdoor triggers, we use a $16 \times 16$ patch with interleaving black and white pixels. In addition to this patch trigger, we evaluate several commonly used triggers in supervised backdoor studies, including Blend with a hello kitty image(Chen et al., 2017), periodical signal pattern (SIG) (Barni et al., 2019), image filter with Nashville style (Liu et al., 2019), and WaNet (Nguyen & Tran, 2021). We also evaluate multiple trigger settings (Li et al., 2024b), where attackers can deploy multiple triggers (MTBA) to target a single entity or multiple entities. Specifically, we use three triggers: the Patch, Nashville style, and WaNet. The multi-trigger settings are denoted as MT-S (single target) and MT-M (multiple targets). Additionally, we investigate the clean-label setting (Turner et al., 2018), where the patch trigger is inserted only into images whose captions already contain the target. An example of trigger patterns used in the experiments is shown in Figure 4.

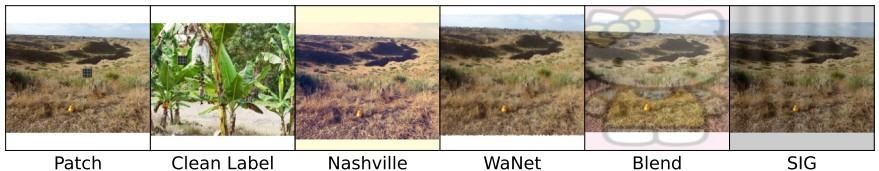

Patch      Clean Label      Nashville      WaNet      Blend      SIG

Figure 4: Examples of the 5 different triggers used in the experiments. The patch and clean label attacks use a 16 by 16 patch as the trigger. The clean label attack only applies the trigger to images with captions that contain the keyword specified by the adversary. The Nashville converts the image using the filter template "Nashville." The WaNet applies grided noise to the image. The SIG uses a periodical pattern as the trigger. The blend attack creates an overly transparent Hello Kitty image.

For the targeted data poisoning attack (TDPA), we randomly select an image and construct the caption set using captions from the training set that contain the target keyword. We chose 'banana' as the target for all attacks because it is an ImageNet class and appears frequently in CC3M.

For backdoor detection, we evaluate local outlier detection and compare it with other backdoor data detection methods. Note that since the existing detection methods are based on supervised learning, we only include the state-of-the-art methods that are feasible in SSL, including ABL (Li et al., 2021) and Cognitive Distillation (CD) (Huang et al., 2023). For CD, we find that in SSL, backdoor data has a higher $L_1$ norm of the mask instead of lower in a supervised setting. As a result, we use a higher $L_1$ norm of the mask to indicate it is more likely to be a backdoor data. We use 100 optimization steps, $\alpha$ set to 0.001 and $\beta$ set to 100 for CD. For ABL, we use the average loss value for the first 10 epochs for each sample. For iForest (Liu et al., 2008), we set the number of trees in the ensemble to 100. For LID, the $k$ is set to 16. For $k$-dist, SLOF and DAO, the $k$ is set to 16. We use batch size of 2048 for running the detection algorithm. For SafeCLIP (Yang et al., 2024), we follow the same hyperparameter setting suggested by the original paper. For warmup training, we perform 5 epochs of uni-modal training followed by 1 epoch of multi-modal training. The learning rate is set to $5 \times 10^{-6}$. The time cost for each detection method on CC3M with the same hardware setting is in Table 3. For all methods, the time cost does not include pre-training time for obtaining the initial model for extracting the representation.

Table 3: Time cost measures the wall time (in hours) for running each detection method on the same hardware. Results are based on ResNet-50 as the vision encoder on CC3M dataset.

| ABL | CD | SafeCLIP | LID | iForest | $k$-dist | SLOF | DAO |
|-----|-----|----------|-----|---------|---------|------|-----|
| 4.1 | 11.2 | 0.2 | 0.2 | 0.3 | 0.2 | 0.2 | 0.2 |

## B.2 SENSITIVITY TO THE LOCALITY

Local outlier methods rely on the locality hyper-parameter $k$ to determine the neighborhood size and could be sensitive to the hyper-parameter. Here, we analyze the sensitivity of different local outlier detection methods to locality $k$ by fixing the batch size to 2048 while testing varying $k$ in $[16, 32, 64, 128, 256]$. Note that setting $k$ too large is not ideal as it will break the assumption of locality. Therefore, we only examine $k$ up to 256. We plot the detection AUC results in Figure 5. It is evident that the detection performance of $k$-dist, SLOF, and DAO is much more stable than LID.

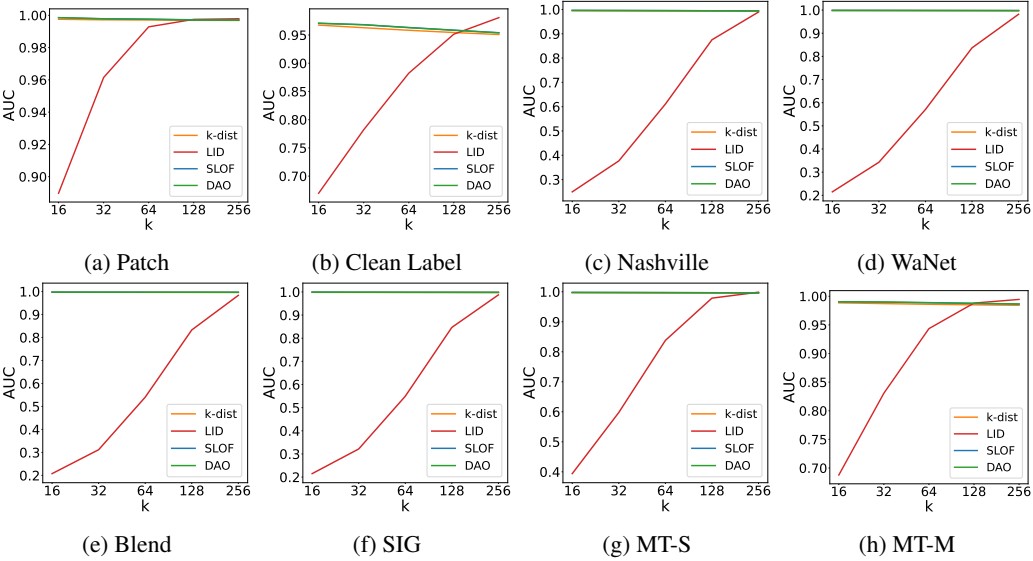

Figure 5: The detection AUC (%) of different local outlier methods under varying locality $k$. The batch size is set to 2048 for all experiments.

## B.3 ADDITIONAL FILTERING RESULTS

We present zero-short evaluation results on 7 commonly used datasets, including CIFAR (Krizhevsky et al., 2009), Food101 (Bossard et al., 2014), GTSRB (Stallkamp et al., 2012), ImageNet (Deng et al., 2009), StanfordCars (Cars) (Krause et al., 2013), STL10 (Coates et al., 2011), in Table 4. We follow the standard zero-shot classification setup and use the template provided by Radford et al. (2021) for each evaluation dataset. We also report the linear probing performance on these datasets in Table 5. Results in Tables 4 and 5 are consistent with Section 5.2. Removing 10% of the samples on CC3M using DAO does not affect the performance of the model.

Table 6 compares filtering-based defense strategies with robust training methods, including RoCLIP (Yang et al., 2023a) and SafeCLIP (Yang et al., 2024). Both methods use unimodal self-supervised objectives alongside the image-text contrastive objective, which is known to enhance CLIP's performance (Li et al., 2022a). For a fair comparison, we also included self-supervised objectives with nearest neighbors (denoted as CLIP+NN) during retraining on the purified subset. As shown in Table 6, retraining CLIP on the purified subset with its original objective effectively mitigates backdoor threats. Incorporating self-supervised objectives further enhances performance and strengthens defenses against backdoor attacks. Note that once potentially poisoned data is removed, any robust retraining method, such as RoCLIP or SafeCLIP, can be applied. The backdoor detection technique can be integrated with other defense strategies.

In Figure 6, we show the CA and ASR with varying filtering percentages. It can be observed that removing different percentages of samples does not significantly affect the CA. Removing 5%–10% samples according to the backdoor scores can effectively mitigate the backdoor threat. In practice, the practitioner could adjust the threshold dynamically to achieve the best performance-security trade-off.

Table 4: Performance of backdoor sample filtering using DAO with filtering rate 10% on poisoned CC3M. The results are presented in the form of clean zero-shot accuracy (%) on the 7 validation set. Results are based on the ResNet-50 as the vision encoder.

| Attack | Dataset | CIFAR10 | CIFAR100 | FOOD101 | GTSRB | ImageNet | Cars | STL10 | Average |
|---|---|---|---|---|---|---|---|---|---|
| BadNets | Poisoned CC3M | 30.7 | 10.7 | 10.5 | 6.2 | 17.0 | 1.4 | 75.4 | 21.7 |
| | Purified CC3M | 32.5 | 10.9 | 10.8 | 9.0 | 16.2 | 1.1 | 74.2 | 22.1 |
| Clean Label | Poisoned CC3M | 36.6 | 12.1 | 11.2 | 6.3 | 17.2 | 1.1 | 68.6 | 21.9 |
| | Purified CC3M | 37.7 | 12.2 | 11.2 | 4.7 | 16.7 | 1.0 | 74.6 | 22.6 |
| Nashville | Poisoned CC3M | 37.4 | 11.6 | 10.4 | 8.2 | 16.7 | 1.3 | 71.9 | 22.5 |
| | Purified CC3M | 36.3 | 12.4 | 11.1 | 6.0 | 16.1 | 1.0 | 67.0 | 21.4 |
| WaNet | Poisoned CC3M | 26.4 | 11.4 | 11.1 | 5.2 | 16.3 | 1.0 | 73.0 | 20.6 |
| | Purified CC3M | 30.5 | 11.0 | 9.8 | 4.8 | 16.7 | 1.1 | 69.3 | 20.5 |
| Blend | Poisoned CC3M | 31.5 | 13.5 | 10.0 | 4.8 | 16.8 | 1.4 | 72.3 | 21.5 |
| | Purified CC3M | 29.3 | 13.4 | 10.2 | 5.3 | 15.9 | 1.0 | 72.0 | 21.0 |
| SIG | Poisoned CC3M | 31.2 | 13.2 | 11.9 | 5.9 | 16.3 | 1.2 | 73.6 | 21.9 |
| | Purified CC3M | 29.4 | 10.8 | 10.1 | 6.5 | 16.4 | 1.0 | 73.0 | 21.0 |
| MT-S | Poisoned CC3M | 36.5 | 11.6 | 10.9 | 4.6 | 16.6 | 1.1 | 72.3 | 21.9 |
| | Purified CC3M | 32.8 | 11.4 | 12.0 | 5.3 | 16.7 | 1.5 | 72.3 | 21.7 |
| MT-M | Poisoned CC3M | 30.6 | 10.9 | 11.4 | 4.7 | 16.2 | 1.0 | 74.0 | 21.2 |
| | Purified CC3M | 32.2 | 12.1 | 12.5 | 8.3 | 16.4 | 1.2 | 72.9 | 22.2 |

Table 5: Performance of backdoor sample filtering using DAO with filtering rate 10% on poisoned CC3M. The results are presented in the form of clean linear prob accuracy (%) on the 7 validation set. Results are based on the ResNet-50 as the vision encoder.

| Attack | Dataset | CIFAR10 | CIFAR100 | FOOD101 | GTSRB | ImageNet | Cars | STL10 | Average |
|---|---|---|---|---|---|---|---|---|---|
| BadNets | Poisoned CC3M | 75.0 | 51.3 | 54.8 | 67.3 | 49.2 | 17.00 | 90.6 | 57.9 |
| | Purified CC3M | 75.5 | 51.5 | 53.4 | 69.4 | 48.3 | 16.6 | 89.1 | 57.7 |
| Clean Label | Poisoned CC3M | 75.0 | 51.5 | 54.5 | 65.9 | 49.1 | 16.5 | 89.8 | 57.4 |
| | Purified CC3M | 74.0 | 51.7 | 54.2 | 69.2 | 48.2 | 16.2 | 89.8 | 57.6 |
| Nashville | Poisoned CC3M | 74.2 | 52.5 | 54.2 | 66.8 | 49.2 | 15.8 | 90.0 | 57.5 |
| | Purified CC3M | 75.2 | 51.2 | 53.8 | 68.0 | 48.2 | 16.7 | 89.7 | 57.5 |
| WaNet | Poisoned CC3M | 75.0 | 51.8 | 54.3 | 67.0 | 49.0 | 16.2 | 90.1 | 57.6 |
| | Purified CC3M | 74.8 | 51.4 | 54.5 | 64.6 | 48.4 | 16.1 | 89.9 | 57.1 |
| Blend | Poisoned CC3M | 75.4 | 51.7 | 54.4 | 66.5 | 49.5 | 16.6 | 90.2 | 57.8 |
| | Purified CC3M | 74.4 | 51.6 | 53.6 | 66.0 | 48.5 | 16.5 | 90.0 | 57.2 |
| SIG | Poisoned CC3M | 75.1 | 51.6 | 54.2 | 67.1 | 49.0 | 16.1 | 90.8 | 57.7 |
| | Purified CC3M | 73.5 | 50.9 | 53.9 | 64.7 | 48.1 | 16.4 | 89.9 | 56.8 |
| MT-S | Poisoned CC3M | 74.2 | 50.0 | 54.4 | 66.4 | 49.2 | 16.6 | 90.0 | 57.3 |
| | Purified CC3M | 68.0 | 48.5 | 47.1 | 63.1 | 46.6 | 16.3 | 90.2 | 54.2 |
| MT-M | Poisoned CC3M | 75.3 | 51.8 | 54.7 | 68.0 | 49.3 | 15.3 | 90.0 | 57.8 |
| | Purified CC3M | 68.9 | 50.8 | 52.3 | 69.7 | 47.7 | 17.5 | 89.7 | 56.7 |

Table 6: Defense performance of backdoor sample filtering using DAO for filtering rate 10% on poisoned CC3M. The results are presented in the form of clean zero-shot accuracy (%) / attack success rate (%) on the ImageNet validation set. Results are based on the ResNet-50 as the vision encoder. The best results in terms of clean zero-shot accuracy and attack success rate are in **boldface**.

| Dataset | Method | Patch | Clean Label | Nashville | WaNet | Blend | SIG | MT-S | MT-M | TDPA |
|---|---|---|---|---|---|---|---|---|---|---|
| Poisoned | CLIP | 17.0 / 100.0 | 17.1 / 95.0 | 16.7 / 78.7 | 16.2 / 83.8 | 16.8 / 75.9 | 16.3 / 67.3 | 16.5 / 79.5 | 16.2 / 74.7 | 16.8 / 100.0 |
| | RoCLIP | 12.8 / 0.1 | 12.7 / **0.0** | 12.5 / 14.6 | 12.6 / 13.4 | 12.2 / 51.4 | 12.3 / 48.2 | 12.8 / 2.0 | 12.2 / 13.9 | 15.2 / 100.0 |
| | SafeCLIP | 17.2 / **0.0** | 17.0 / 19.9 | 16.7 / 54.5 | 17.4 / 9.4 | **17.6** / 53.6 | 16.5 / 68.7 | 16.8 / 32.6 | 17.1 / 30.9 | 17.2 / 100.0 |
| Purified (Ours) | CLIP | 16.2 / **0.0** | 16.7 / **0.0** | 16.1 / 9.6 | 16.7 / 0.5 | 15.8 / 0.6 | 16.4 / 0.2 | 16.7 / 0.6 | 16.4 / 15.00 | 16.3 / **0.0** |
| | CLIP+NN | **17.4** / **0.0** | **17.5** / **0.0** | **17.4** / **0.3** | **17.6** / **0.1** | 17.5 / **0.0** | **17.6** / **0.1** | **17.9** / **0.1** | **17.6** / **0.2** | **17.7** / **0.0** |

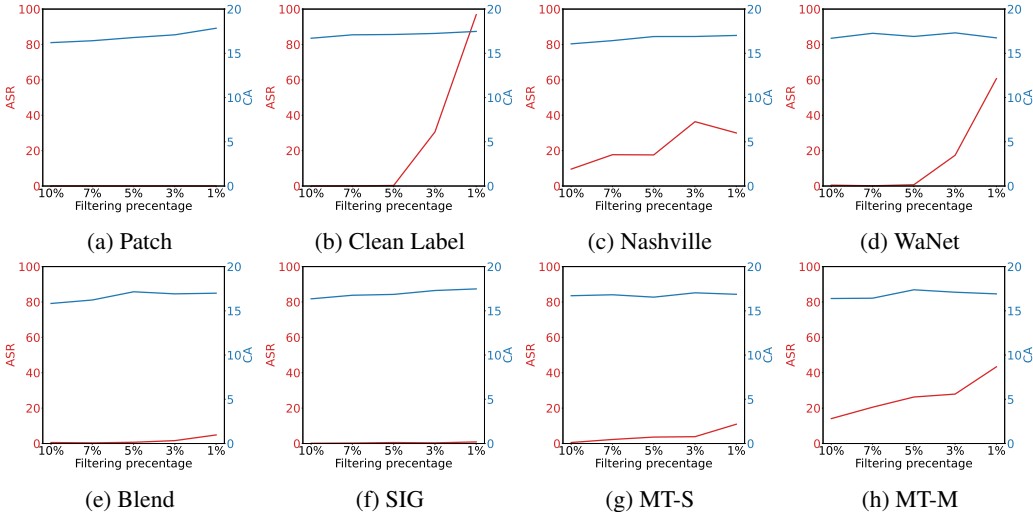

(a) Patch      (b) Clean Label      (c) Nashville      (d) WaNet

(e) Blend      (f) SIG      (g) MT-S      (h) MT-M

Figure 6: The attack success rate (ASR) and clean accuracy (CA) were evaluated using zero-shot classifications on ImageNet with varying filtering percentages. Results are based on ResNet-50 as the vision encoder.

### B.4 TRIGGER SYNTHESIS

In this subsection, we present the trigger synthesis method we used to recover the trigger on the birthday cake images that are presented in Section 5.3. Since existing methods are designed for supervised learning, we need to make adaptations in order to use them for CLIP. Inspired by Neural Cleanse (Wang et al., 2019) and Cognitive Distillation (Huang et al., 2023), we use the following objective:

$$\arg\min_{\boldsymbol{m},\boldsymbol{\Delta}} \text{sim}(f_I(\boldsymbol{x}'), \boldsymbol{z}^t) + \alpha\|\boldsymbol{m}\|_1 + \beta(TV(\boldsymbol{m}) + TV(\boldsymbol{\Delta})) \tag{2}$$

$$\boldsymbol{x}' = \boldsymbol{m} \odot \Delta + (1 - \boldsymbol{m}) \odot \boldsymbol{x}, \tag{3}$$

where $\boldsymbol{m} \in [0,1]^{w \times h}$ is a learnable 2D input mask that does not include the color channels, $\Delta \in [0,1]^{3 \times w \times h}$ is the trigger pattern, $\odot$ is the element-wise multiplication applied to all the channels, $TV(\cdot)$ is the total variation loss, $\boldsymbol{z}^t = f_T(t)$ is the embedding of the target caption, $f_I$ is the image encoder and $\text{sim}(\cdot)$ is the similarity measure.

For the birthday cake images, the target caption is "*the birthday cake with candles in the form of number icon.*" We perform the trigger synthesis using the equations above on the CC3M dataset and run for optimization 250 steps, $\alpha$ is set to 0.0001, and $\beta$ to 70. We use Adam (Kingma & Ba, 2014) as the optimizer for $\boldsymbol{m}$ and $\boldsymbol{\Delta}$, the learning rate is set to 0.05, $\beta_1$ and $\beta_2$ are set to 0.1. $\boldsymbol{m}$ is initialized using 1, and $\boldsymbol{\Delta}$ is initialized using a birthday cake example.

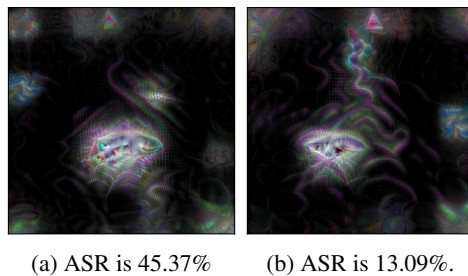

(a) ASR is 45.37%      (b) ASR is 13.09%.

Figure 7: (a-b) The synthesized patterns with "a photo of great white shark" as the target caption.

The trigger synthesis might appear to be similar to a targeted universal adversarial attack and might not validate that the birthday cake is a real backdoor. To address this, we provide a counter-example.

This trigger synthesis is not effective in creating a strong targeted-universal adversarial attack. We use one of the ImageNet classes as the target and the prompt template "a photo of {target}" as the target caption. We repeat the experiment presented in the main paper with the exact same hyperparameters except the $\Delta$ initialized with random values. We conducted this experiment twice. The recovered triggers are shown in Figure 7. These triggers only achieve an ASR of 45.37% and 13.09%. Not as high as the birthday cake example (92.38%). The high ASR of the birthday cake example makes it highly susceptible to being a backdoor trigger.

### B.5 Sensitivity to Poisoning Rates

In this subsection, we examine the sensitivity of local outlier detection methods to varying poisoning rates. The results are presented in Table 7. With our default setting of $k = 16$, $k$-dist shows the most robustness against changes in the poisoning rate. SLOF and DAO remain relatively stable up to a 5% poisoning rate. When the poisoning rate significantly increases to 10%, increasing the locality $k$ to 256 substantially improves performance compared to $k = 16$. This is expected since a higher poisoning rate increases the likelihood that the $k$ nearest neighbors will include poisoned samples, necessitating an adjustment in the locality $k$. We present the embedding space visualization with the control experiments to increase the number of backdoor samples within the batch in Figure 8. It can be observed that with an increase in the locality parameter $k$, the local outlier methods can accurately identify backdoor samples when the poisoning rate is significantly higher.

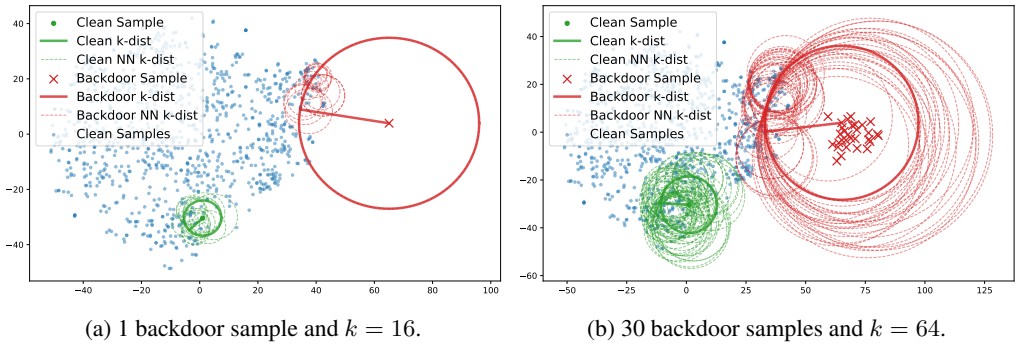

(a) 1 backdoor sample and $k = 16$.      (b) 30 backdoor samples and $k = 64$.

Figure 8: The t-SNE plot of the embedding space with various numbers of backdoor samples for a batch of 1024 data points.

In real-world web-scale datasets, poisoning more than 5% of the data is highly unlikely. Carlini et al. (2024) indicates that while poisoning 0.01% of a dataset is relatively inexpensive, poisoning 0.1% would drastically increase the cost from \$10 USD to \$10,000 USD. Poisoning 5% to 10% of a web-scale dataset would be extremely costly. Therefore, we believe that the default choice of $k$=16 is appropriate. However, using $k$=256 as a default is also feasible, as detection performance remains stable for $k$ between 16 and 256, as demonstrated in Appendix B.2. Unlike SafeCLIP (Yang et al., 2024), which identified difficulties in detecting backdoor samples when the poisoning rate exceeds 0.5%, local outlier detection remains robust to poisoning rates as high as 10%, although such scenarios are unrealistic in practice.

Table 7: Results of sensitivity towards poisoning rate for local methods. All results are based on the Patch trigger. Results are reported as area under the ROC curve. The default $k$ is 16.

| Poisoning Rate | LID | $k$-dist | SLOF | DAO |
|---|---|---|---|---|
| 0.01% | 99.29 | 99.75 | 99.86 | 99.86 |
| 0.1% | 69.82 | 100.00 | 100.00 | 100.00 |
| 1% | 0.29 | 100.00 | 100.00 | 100.00 |
| 5% | 0.01 | 100.00 | 99.87 | 99.66 |
| 10% | 1.18 | 95.39 | 63.00 | 64.88 |
| 10% ($k$=256) | 0.00 | 100.00 | 99.99 | 99.98 |

LID is shown to be sensitive to poisoning rates. Interestingly, in the case of a 10% poisoning rate, LID detection shows an AUC of 0.0, suggesting that all poisoned samples have a low LID score at higher poisoning rates. In contrast, at lower poisoning rates, poisoned samples tend to have higher LID scores. This indicates that LID detection is not robust to changes in the poisoning rate.

## B.6 WHITE-BOX ADAPTIVE ATTACKS

In this subsection, we provide an analysis of local outlier detection methods against white-box adaptive attacks, i.e., the attacker is aware of our detection strategy and attempts to evade our detection. We assume the attacker can control the training process to regularize the outlier scores so that the trigger is small. This is the unrealistic setting for data poisoning attacks, but such analysis could provide insights into the robustness of local outlier detection methods.

To evade the detection, the attacker may add a regularization term to the original training objective, forcing the model to generate smaller outliers for the backdoor samples. Formally, it is defined as the following:

$$\mathcal{L}_{\text{CLIP}}(\mathbf{z}^x, \mathbf{z}^t) + \text{SLOF}(\mathbf{z}^{x'}), \quad \mathbf{x}' \in \mathcal{D}_b,$$

where $\mathbf{z}^x$ and $\mathbf{z}^t$ are the image and text embeddings, respectively, and $\text{SLOF}(\mathbf{z}^{x'})$ denotes the Local Outlier Factor score of the backdoor-poisoned samples $\mathbf{x}'$ in the backdoor dataset $\mathcal{D}_b$. This objective allows the attacker to minimize the outlier score of the poisoned samples during pretraining.

Table 8: Comparing the AUC (%) of different outlier detection methods against adaptive attacks. Clean Acc (CA) and ASR are measured by the top-1 zero-shot accuracy (%) on ImageNet. The best results are **boldfaced**.

| Method | Trigger | Poisoning Rate | Clean Acc | ASR | $k$-dist | SLOF | DAO |
|---|---|---|---|---|---|---|---|
| Standard | Patch | 0.01% | 17.00 | 100.0 | 99.75 | **99.86** | **99.86** |
| Adaptive Attack | Patch | 0.01% | 15.94 | 100.0 | **100.0** | **100.0** | **100.0** |

The results are reported in the table 8. It clearly shows that this adaptive strategy does not circumvent our detection method; in fact, it even improves our detection performance. This is because forcing the poisoned backdoor samples to mimic the density profile of clean samples is only effective within a specific neighborhood in the feature space. To fully evade detection, an attacker would need to account for all possible neighborhoods generated by various combinations of data points—a task that is computationally infeasible given the scale of web datasets, which often contain millions or even billions of samples. Therefore, our detection method remains robust even against adaptive attacks that attempt to minimize the outlier scores of poisoned samples. This reinforces the effectiveness of our approach in real-world settings where attackers may employ sophisticated strategies to hide backdoor triggers.

## B.7 ADDITIONAL DETECTION RESULTS

In this section, we present the result of FPR@95 in Table 9, which is the false positive rate at 95% true positive rate. This is a complementary metric to the AUC score. In practice, it's difficult to set a threshold that achieves exactly a 95% true positive rate when removing backdoor data. Therefore, AUC, which represents the probability of a backdoor sample having a higher score than a clean sample, is the preferred metric for detecting backdoor samples. Nevertheless, we report the FPR@95 here. The results are consistent with Section 5.1 of the main paper, where local outlier methods consistently perform well. When using RN50 as the encoder, local outlier methods only show an FPR@95 above 1% for Clean Label and MT-M attacks. As demonstrated in Section 5.3, even with clean datasets, these methods detect noisy data (counted as false positives) that do not benefit pretraining. A slightly higher FPR does not impact clean data performance during removal and retraining, as demonstrated in Section 5.2 and Appendix B.3.

We present the detection results for the CC12M (Changpinyo et al., 2021) and RedCaps (Desai et al., 2021) dataset with RN50 as an image encoder in Table 10 and 11. It can be observed the local outlier detection consistently outperforms other baselines. The findings are consistent as in Section 5.1 in the main paper.

We provide extended results on the distributions of backdoor scores using ABL, CD, LID, $k$-dist, Isolation Forest, and SLOF in Figures 9–16. Results are based on using RN50 as the image encoder.

Table 9: The detection performance is evaluated using the FPR@95, and results are reported with percentage (%). The best results are **boldfaced**.

| Vision Encoder | Threat Model | Trigger | Poisoning Rate (%) | CA (%) | ASR (%) | ABL | CD | Safe CLIP | LID | iForest | $k$-dist | SLOF | DAO |
|---|---|---|---|---|---|---|---|---|---|---|---|---|---|
| RN50 | STBA | Patch | 0.01 | 17.0 | 100.0 | 99.48 | 13.67 | 52.62 | 3.06 | 0.44 | 0.32 | **0.25** | 0.28 |
| | | Clean Label | 0.07 | 17.1 | 95.0 | 87.06 | 95.55 | 98.03 | 50.81 | 23.83 | 12.75 | **11.23** | 11.45 |
| | | Nashville | 0.1 | 16.7 | 78.7 | 87.81 | **0.24** | 84.18 | 97.98 | 0.36 | 0.29 | 0.27 | 0.27 |
| | | WaNet | 0.1 | 16.2 | 83.8 | 89.78 | 1.88 | 46.54 | 98.82 | 0.74 | 0.26 | **0.30** | 0.32 |
| | | Blend | 0.1 | 16.8 | 75.9 | 90.01 | **0.0** | 74.06 | 99.51 | 0.03 | 0.03 | 0.03 | 0.03 |
| | | SIG | 0.1 | 16.3 | 67.3 | 92.01 | 0.07 | 49.23 | 99.59 | 0.07 | **0.03** | 0.04 | 0.05 |
| | MTBA | MT-S | 0.1 | 16.5 | 79.5 | 94.69 | 0.51 | 50.65 | 36.80 | 0.28 | 0.23 | 0.21 | **0.20** |
| | | MT-M | 0.1 | 16.2 | 74.7 | 93.08 | 25.85 | 51.83 | 16.92 | 6.42 | 5.85 | 4.36 | **4.30** |
| | TDPA | - | 0.01 | 16.8 | 100.0 | 89.25 | **0.01** | 60.42 | 99.97 | 9.46 | 0.05 | 0.06 | 0.06 |
| ViT B-16 | STBA | Patch | 0.1 | 15.2 | 99.8 | 99.02 | 99.04 | 38.49 | 98.63 | 48.78 | **6.23** | 22.91 | 30.33 |
| | | Clean Label | 0.07 | 15.7 | 19.0 | 91.39 | 89.09 | 97.47 | **79.07** | 92.92 | 84.17 | 79.08 | 79.76 |
| | | Nashville | 0.1 | 15.7 | 41.4 | 90.52 | 25.18 | 69.41 | 99.90 | 34.64 | **1.23** | 12.77 | 20.04 |
| | | WaNet | 0.1 | 15.2 | 12.2 | 98.38 | 85.09 | 43.41 | 99.67 | 44.67 | **4.83** | 22.06 | 32.54 |
| | | Blend | 0.1 | 15.7 | 95.8 | 84.61 | 7.09 | 77.21 | 99.94 | 42.71 | **0.07** | 62.72 | 71.68 |
| | | SIG | 0.1 | 15.3 | 82.9 | 91.64 | 99.79 | 70.13 | 99.94 | 42.00 | **0.15** | 13.00 | 23.89 |
| | MTBA | MT-S | 0.1 | 15.3 | 28.5 | 99.74 | 93.95 | 49.03 | 90.01 | 53.58 | 28.40 | 21.60 | **20.80** |
| | | MT-M | 0.1 | 15.2 | 36.2 | 99.62 | 92.25 | 57.23 | 81.13 | 79.04 | 68.86 | 56.07 | **55.10** |
| | TDPA | - | 0.01 | 15.5 | 100.0 | 86.63 | 72.59 | 32.53 | 99.98 | **0.05** | 6.10 | 5.72 | 6.82 |

Table 10: Results for CC12M dataset. The detection performance is evaluated using the AUC, and results are reported with percentage (%). The best results are **boldfaced**.

| Threat Model | Trigger | Poisoning Rate (%) | CA (%) | ASR (%) | ABL | CD | Safe CLIP | LID | iForest | $k$-dist | SLOF | DAO |
|---|---|---|---|---|---|---|---|---|---|---|---|---|
| STBA | Patch | 0.01 | 27.1 | 100.0 | 48.02 | 98.34 | 88.70 | 99.27 | 99.53 | 99.37 | 99.50 | **99.54** |
| | Clean Label | 0.07 | 27.7 | 91.5 | 62.98 | 78.29 | 40.88 | 65.12 | 86.72 | 87.86 | **89.73** | 89.66 |
| | Nashville | 0.1 | 20.5 | 99.3 | 56.63 | 99.07 | 25.87 | 58.66 | 99.84 | 99.72 | **99.85** | **99.85** |
| | WaNet | 0.1 | 24.8 | 92.3 | 57.66 | 98.52 | 88.34 | 52.42 | 99.62 | 99.50 | 99.66 | **99.69** |
| | Blend | 0.1 | 26.8 | 96.0 | 58.97 | 99.65 | 30.32 | 48.46 | 99.81 | 99.82 | **99.83** | **99.83** |
| | SIG | 0.1 | 26.1 | 61.7 | 58.65 | 99.36 | 58.14 | 49.94 | 99.61 | 99.63 | **99.67** | 99.66 |
| MTBA | MT-S | 0.1 | 14.5 | 99.4 | 50.82 | 99.56 | 84.81 | 80.49 | 99.80 | 99.78 | **99.84** | **99.84** |
| | MT-M | 0.1 | 14.7 | 98.9 | 57.43 | 97.47 | 74.03 | 95.24 | 99.02 | 99.15 | 99.30 | **99.33** |
| TDPA | - | 0.01 | 26.7 | 100.0 | 46.76 | 99.60 | 86.03 | 80.69 | 99.98 | **100.0** | 99.98 | 99.98 |

Table 11: Results for RedCaps dataset. The detection performance is evaluated using the AUC, and results are reported with percentage (%). The best results are **boldfaced**.

| Threat Model | Trigger | Poisoning Rate (%) | CA (%) | ASR (%) | ABL | CD | Safe CLIP | LID | iForest | $k$-dist | SLOF | DAO |
|---|---|---|---|---|---|---|---|---|---|---|---|---|
| STBA | Patch | 0.01 | 29.9 | 96.9 | 41.01 | 93.27 | 78.05 | 93.93 | 93.55 | 93.56 | 93.74 | **93.75** |
| | Clean Label | 0.07 | 30.4 | 94.9 | 70.15 | 57.04 | 44.66 | 78.87 | 85.41 | 87.58 | **89.25** | 89.07 |
| | Nashville | 0.1 | 29.7 | 85.4 | 69.28 | 98.23 | 6.22 | 69.26 | 99.16 | 99.29 | **99.40** | 99.38 |
| | WaNet | 0.1 | 28.0 | 35.8 | 64.83 | 96.53 | 76.53 | 54.3 | 99.78 | 99.84 | **99.89** | 99.88 |
| | Blend | 0.1 | 30.0 | 98.3 | 70.34 | 99.55 | 35.57 | 61.14 | 99.86 | 99.88 | **99.93** | **99.93** |
| | SIG | 0.1 | 29.5 | 97.9 | 69.35 | 93.57 | 68.76 | 69.48 | 99.89 | 99.90 | **99.91** | **99.91** |
| MTBA | MT-S | 0.1 | 29.5 | 83.0 | 61.16 | 95.92 | 63.33 | 86.71 | 97.37 | 97.55 | **97.73** | **97.73** |
| | MT-M | 0.1 | 27.7 | 83.9 | 63.43 | 90.67 | 58.98 | 94.51 | 96.38 | 96.71 | **97.04** | 97.02 |
| TDPA | - | 0.01 | 30.0 | 100.0 | 51.71 | 99.93 | 89.66 | 81.83 | 99.71 | 99.88 | 99.95 | **99.96** |

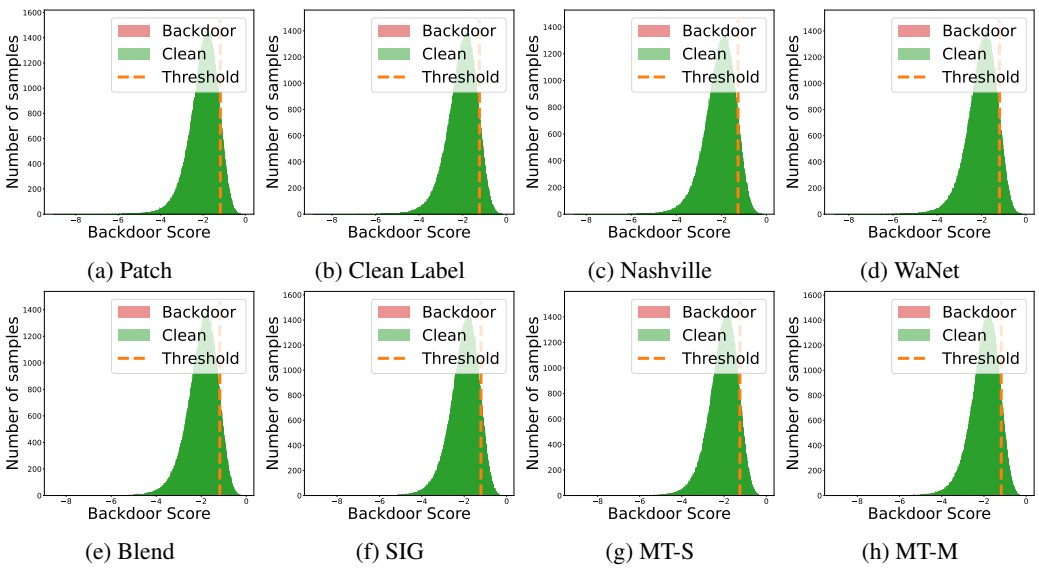

Figure 9: The distribution of backdoor scores using ABL on the CC3M dataset.

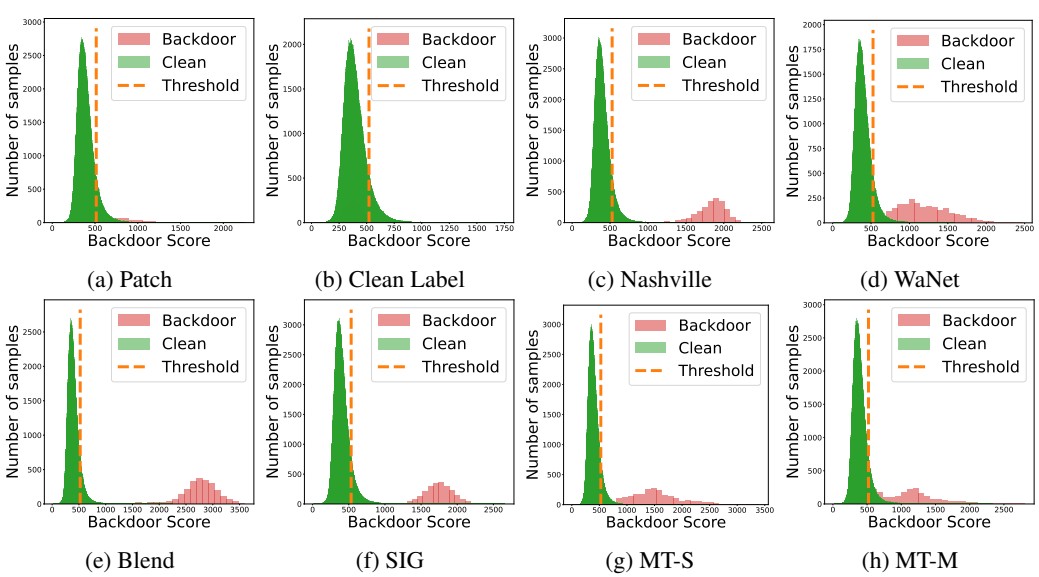

Figure 10: The distribution of backdoor scores using CD on the CC3M dataset.

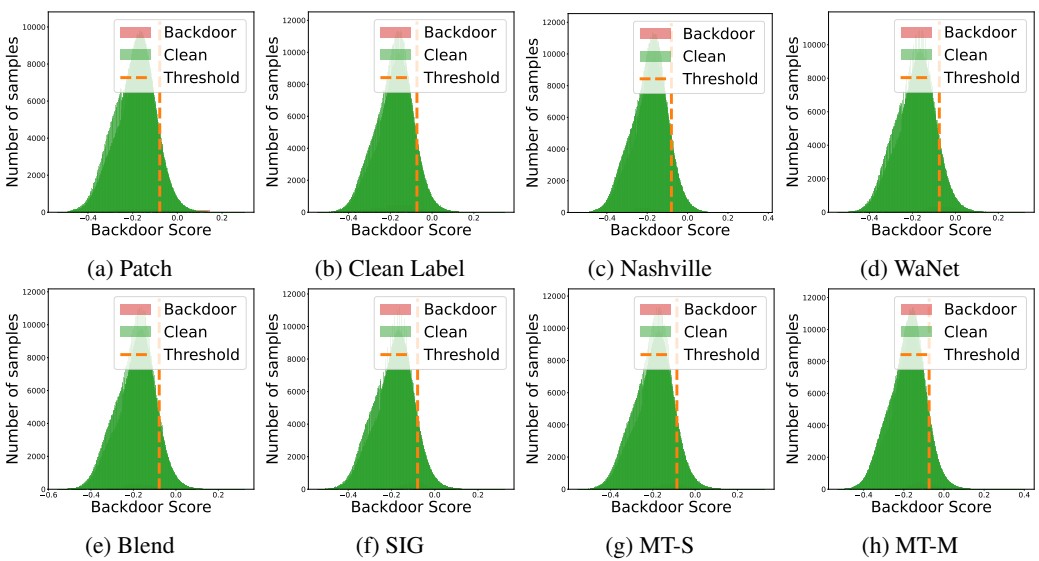

Figure 11: The distribution of backdoor scores using SafeCLIP on the CC3M dataset.

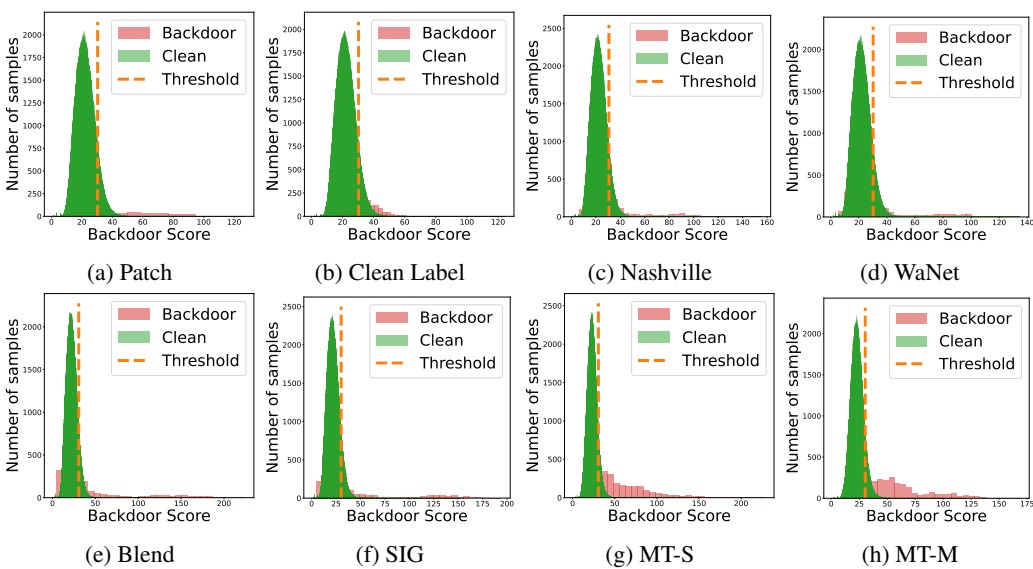

Figure 12: The distribution of backdoor scores using LID on the CC3M dataset.

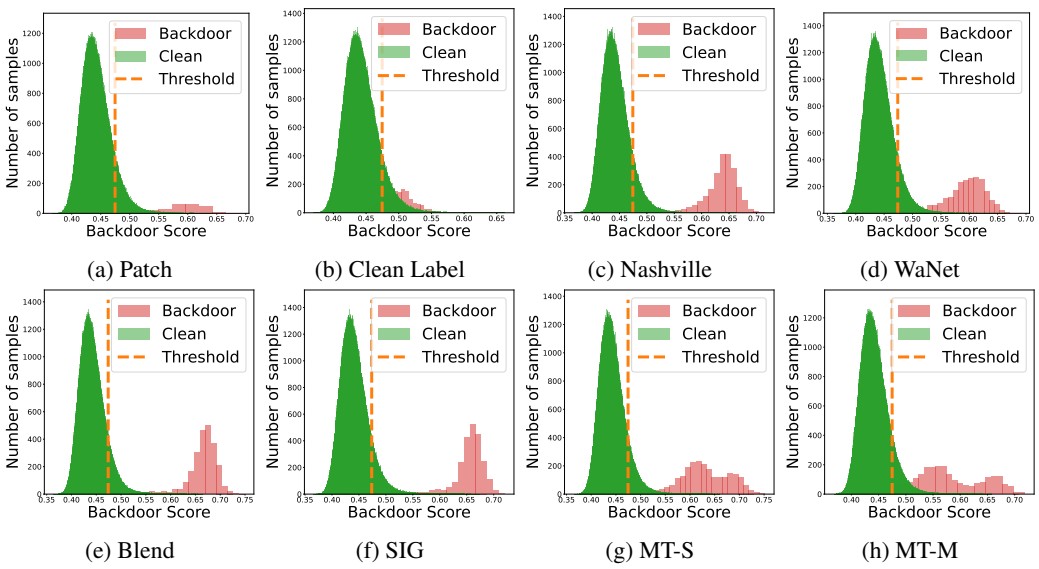

Figure 13: The distribution of backdoor scores using iForest on the CC3M dataset.

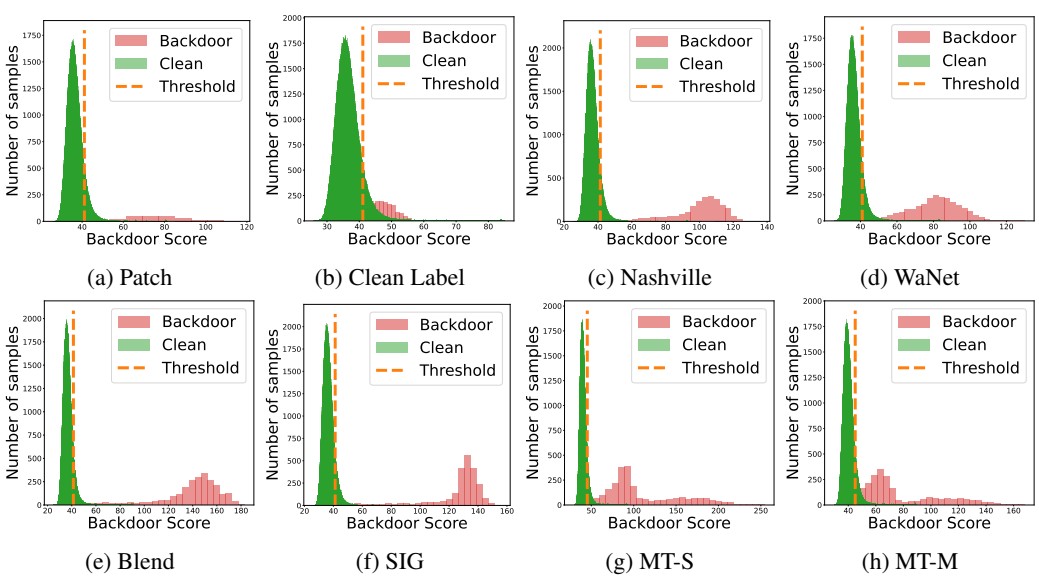

Figure 14: The distribution of backdoor scores using $k$-dist on the CC3M dataset.

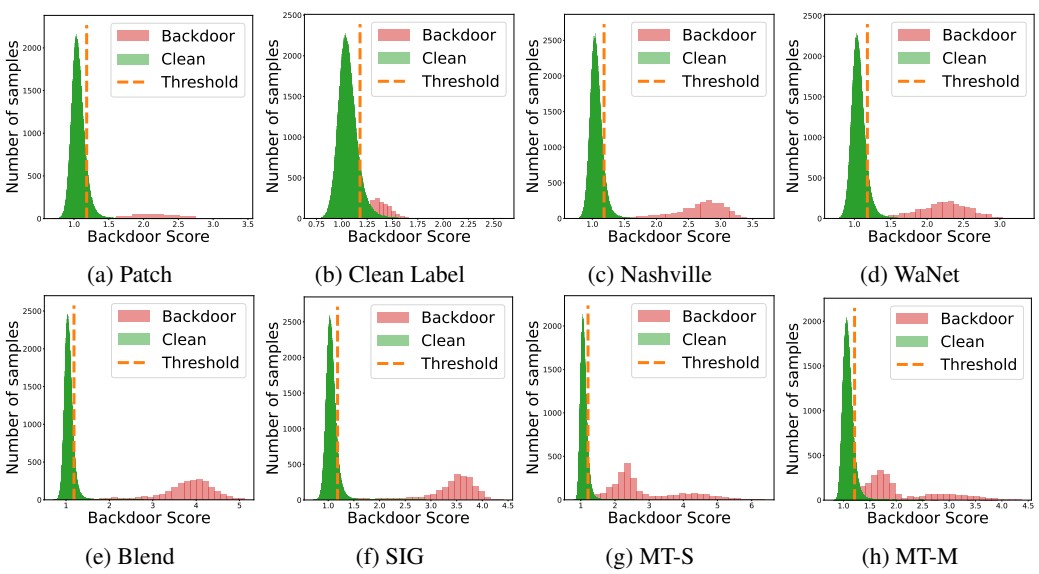

Figure 15: The distribution of backdoor scores using SLOF on the CC3M dataset.

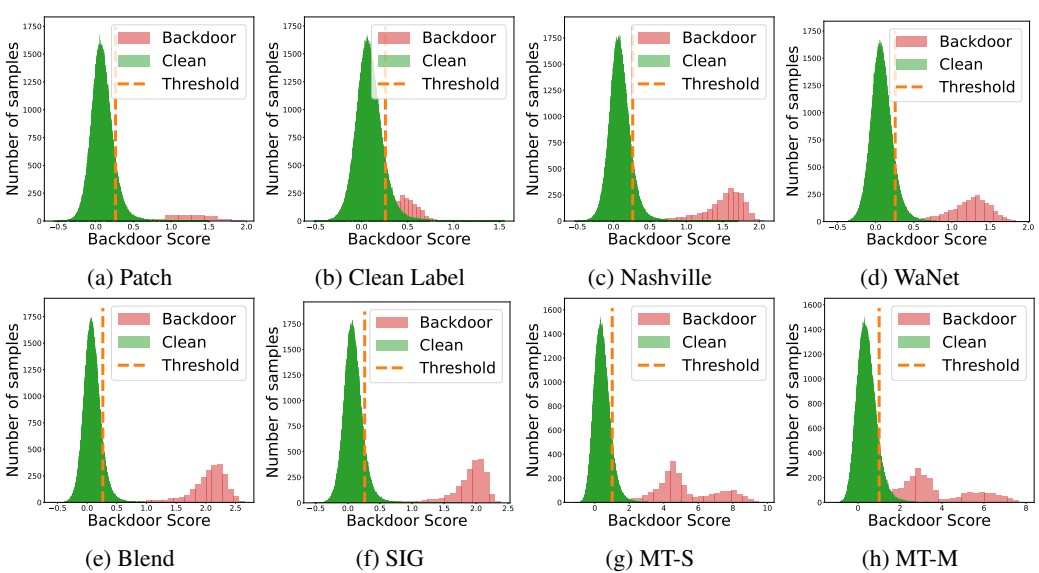

Figure 16: The distribution of backdoor scores using DAO on the CC3M dataset.

