# OpenReview forum: "Detecting Backdoor Samples in Contrastive Language Image Pretraining"
_ICLR.cc/2025/Conference — ICLR 2025 Poster_

### Official Review · Reviewer_5XDB · 2024-10-29

**Soundness:** 3
**Presentation:** 3
**Contribution:** 3
**Rating:** 6
**Confidence:** 4

**Summary:**

This paper focuses on the backdoor attacks on CLIP models. The researchers identify unique characteristics of poisoned samples and develop an efficient detection method using local outlier detectors. They discover an unintentional backdoor in the CC3M dataset and provide a fast solution to clean large-scale datasets, demonstrating its effectiveness by processing CC3M in just 15 minutes using 4 GPUs. This work highlights the importance of scrutinizing training data for large-scale AI models and offers a practical approach to enhance their security.

**Strengths:**

1. The paper focuses on a compelling and timely topic in AI security.
2. A key discovery is that backdoor-poisoned samples in CLIP models exhibit distinctive characteristics in their local subspace, notably sparser local neighborhoods compared to clean samples.
3. The research reveals an intriguing and previously unknown unintentional backdoor in the widely-used CC3M dataset.

**Weaknesses:**

1. The core defense strategy relies on the observation that backdoor examples exhibit sparser local neighborhoods compared to clean samples. This approach is particularly effective when the poisoning ratio is low, as the k-nearest neighbors of a backdoor example are likely to be clean samples. However, if my understanding is correct, as the poisoning ratio increases, the selection of the hyperparameter k becomes crucial. How to choose the hyperparameters of detection algorithms?
2. As claimed by the authors,  all the backdoor-poisoned samples contain similar features (the trigger) and are likely to be clustered together in a particular region. Intuitively, these backdoor examples will form a denser region. Besides, according to a previous study [1], the backdoor example region is much denser compared to clean examples.  However, in this paper, the authors claim that backdoor examples exhibit sparser local neighborhoods. How to explain such a difference?
3. Lacking discussion of adaptive attacks.



[1] Li, C., Pang, R., Cao, B., Xi, Z., Chen, J., Ji, S., & Wang, T. (2024). On the Difficulty of Defending Contrastive Learning against Backdoor Attacks. In 33rd USENIX Security Symposium (USENIX Security 24) (pp. 2901-2918).

**Questions:**

Please see comments.

---

> ### Author Response · Authors · 2024-11-19
> **Response to Reviewer 5XDB (1/2)**
>
> Thank you for taking the time to review our paper and for your valuable comments. Please find our responses to your questions below.
>
> ---
>
> **Q1:** Impact of k and choice of this hyperparameter.
>
> **A1:** As demonstrated in Appendix B.5, our detection method remains effective even with a 10% poisoning rate, a level that is highly unlikely in practice according to existing studies.
>
> > How to choose the hyperparameters of detection algorithms?
>
> Appendix B.5 provides guidance on selecting the hyperparameter $\( k \)$. It suggests that the ratio $\( \frac{k}{b} \)$ (where $\( b \)$ is the batch size) should exceed the poisoning rate. In our experiments, the default batch size is 2048. For instance:
>
> - In Appendix B.5, where $\( k = 256 \)$, the ratio $\( \frac{k}{b} = 12.5$%$\) $.
> - In Table 1, $\( \frac{k}{b} = 0.78 $%$ \)$.
>
> If the ratio $\( \frac{k}{b} \)$ is less than the poisoning rate, it is possible that all $\( k \)$ neighbors are poisoned data, rendering the outlier scores meaningless. Therefore, we recommend setting $\( \frac{k}{b} \)$ above the expected poisoning rate.
>
> However, in real-world scenarios, the defender does not know the true poisoning rate. In this case, we believe $\( k = 16 \) $ is a reasonable choice, given the cost of poisoning the dataset in practice. Furthermore, setting a larger $\( k \)$ (and hence a higher $\( \frac{k}{b} \)$ ratio) has minimal impact on detection performance.
>
> Below, we present the detection results for $\( k = 256 \)$ $(\( \frac{k}{b} = 12.5\$%$ \))$ under poisoning rates of 0.01% to 0.1%, illustrating the robustness of our method across varying $\( k \)$ values.
>
>
> | Trigger | ABL | CD | SafeCLIP | LID | iForest | DAO k=16 | DAO k=32 | DAO k=64 | DAO k=128 | DAO k=256 |
> |:---:|:---:|:---:|:---:|:---:|:---:|:---:|:---:|:---:|:---:|:---:|
> | Patch | 27.86 | 97.17 | 83.42 | 99.29 | `99.73` | **99.86** | 99.78 | 99.75 | 99.71 | `99.70` |
> | Clean Label | 63.50 | 48.68 | 46.93 | 88.23 | `94.01` | **97.06** | 96.81 | 96.30 | 95.83 | `95.38` |
> | Nashville | 61.69 | 98.33 | 46.37 | 61.07 | `99.35` | **99.62** | 99.59 | 99.54 | 99.49 | `99.44` |
> | WaNet | 56.07 | 99.19 | 85.82 | 57.07 | `99.55` | **99.85** | 99.84 | 99.82 | 99.80 | `99.77` |
> | Blend | 60.07 | 99.64 | 57.01 | 54.94 | `99.80` | **99.88** | 99.86 | 99.84 | 99.82 | `99.80` |
> | SIG | 56.88 | 99.06 | 82.15 | 54.03 | `99.62` | **99.69** | 99.68 | 99.67 | 99.66 | `99.64` |
> | MT-S | 47.32 | 99.34 | 80.98 | 94.87 | `99.59` | **99.77** | 99.75 | 99.71 | 99.68 | `99.63` |
> | MT-M | 53.24 | 95.33 | 78.10 | 97.11 | `98.32` | **99.04** | 99.00 | 98.90 | 98.77 | `98.66` |
>
> ---
>
> ---
>
> **Q2:** Dense backdoor region demonstrated by existing work.
>
> **A2:**
>
> > according to a previous study [1], the backdoor example region is much denser compared to clean examples.
>
> The density of backdoor examples depends on the context and granularity of the analysis. The method referenced by the reviewer evaluates the entire dataset and calculates class-wise distances, where samples within the same class (including backdoor-poisoned samples) are likely to form dense clusters. In contrast, our approach examines the representation space within a randomly sampled batch, without relying on any label information. This means that our method does not assess class-wise distances, resulting in a different neighborhood context and potentially different density observations.
>
> > all the backdoor-poisoned samples contain similar features (the trigger) and are likely to be clustered together in a particular region.
>
> In our controlled experiments, as shown in Figure 1(b), we observe that backdoor data do indeed cluster together in the representation space due to shared features like the trigger. However, when clean samples are included in the neighborhood, the backdoor data become relatively sparse compared to the clean samples. This highlights that density measures must be interpreted within the specific context of the neighborhood.
> It is also worth emphasizing that the outlier score (e.g., SLOF or DAO) used in our method is based on the density ratio, not absolute density. This distinction is crucial: backdoor samples may cluster tightly together in isolation, but their sparsity relative to clean samples within the local neighborhood makes them identifiable as outliers. This observation directly relates to Q1/A1, further supporting our method's ability to detect backdoor samples by leveraging the contrast between clean and backdoor samples in the local neighborhood.

---

> ### Author Response · Authors · 2024-11-19
> **Response to Reviewer 5XDB (2/2)**
>
> **Q3:** Analysis with adaptive attack
>
> **A3:**
>
> Thank you for your thoughtful suggestion. We have conducted additional experiments to analyze adaptive attacks that aim to circumvent our detection method. Specifically, we tested an adaptive attack using the following optimization objective during pretraining:
>
> $$
> \mathcal{L}_{\text{CLIP}}(\mathbf{z}^x, \mathbf{z}^t) + \text{SLOF}(\mathbf{z}^{x'}), \quad \mathbf{x}' \in \mathcal{D}_b,
> $$
>
> where $\mathbf{z}^x$ and $\mathbf{z}^t$ are the image and text embeddings, respectively, and $\text{SLOF}(\mathbf{z}^{x'})$ denotes the Local Outlier Factor score of the backdoor-poisoned samples $\mathbf{x}'$ in the backdoor dataset $\mathcal{D}_b$. This objective allows the attacker to minimize the outlier score of the poisoned samples during pretraining. We implemented this adaptive attack using the **BadNets** patch trigger, keeping all other experimental settings consistent with those in our initial submission.
>
> The results are reported in the table below. It clearly shows that this adaptive strategy does not circumvent our detection method; in fact, it even improves our detection performance. This is because forcing the poisoned backdoor samples to mimic the density profile of clean samples is only effective within a specific neighborhood in the feature space. To fully evade detection, an attacker would need to account for all possible neighborhoods generated by various combinations of data points—a task that is computationally infeasible given the scale of web datasets, which often contain millions or even billions of samples. Therefore, our detection method remains robust even against adaptive attacks that attempt to minimize the outlier scores of poisoned samples. This reinforces the effectiveness of our approach in real-world settings where attackers may employ sophisticated strategies to hide backdoor triggers.
>
> |      Method     | Trigger | Poisoning Rate | Clean Acc |  ASR  |  $k$-dist |    SLOF   |    DAO    |
> |:---------------:|:-------:|:--------------:|:---------:|:-----:|:---------:|:---------:|:---------:|
> |     Standard    |  Patch  |      0.01%     |   17.00   | 100.0 |   99.75   | **99.86** | **99.86** |
> | Adaptive Attack |  Patch  |      0.01%     |   15.94   | 100.0 | **100.0** | **100.0** | **100.0** |

---

> > ### Comment · Reviewer_5XDB · 2024-11-22
> > **Reply to authors**
> >
> > 1. Are the results from the table evaluated under poisoning rates of 0.01% to 0.1%? How do you read the variation in the results under different poisoning rates?  Or just evaluated under a fixed rate? I can only know that K (from 16 - 256) does not influence the effectiveness.
> >
> > 2. From my understanding, this is not related to label information. Once poisoned, the feature distribution is there. For example, if you randomly sample 1000 examples, you will get around 100 poisoned examples under the poison ratio of 10%. These 100 examples will cluster together densely due to the common trigger feature. In this setting, do you mean the poisoned examples have higher density but lower density ratio? If it is, how to explain this?
> >
> > 3. Does Including an additional SLOF loss term result in a higher SLOF, this doesn't make sense. If the two items have a tradeoff, we will observe the SLOF decrease but a higher CLIP loss; if they don't have a tradeoff, we will observe both items decrease. This is counterintuitive. How to explain this?

---

> > > ### Author Response · Authors · 2024-11-23
> > > **Response to Reviewer 5XDB**
> > >
> > > We appreciate the reviewer's insightful follow-up questions. Please find the explanation below.
> > >
> > > ---
> > >
> > > **Q1:** K (from 16 - 256) does not influence the effectiveness.
> > >
> > > **A1:**
> > >
> > > > Are the results from the table evaluated under poisoning rates of 0.01% to 0.1%? How do you read the variation in the results under different poisoning rates? Or just evaluated under a fixed rate?
> > >
> > > The results presented in "Response to Reviewer 5XDB (1/2)" are consistent with those in Table 1 of our initial submission. For clarity and ease of comparison, we have consolidated the detection results from Figure 5 and Table 1. All detection methods were evaluated using the same fixed poisoned subset. Specifically, for DAO with *k* varying from 16 to 256, we utilized the identical poisoned subset, applying a poisoning rate of 0.01% for patch attacks, 0.07% for clean-label attacks, and 0.1% for all other attack types.
> > >
> > > Our findings indicate that varying *k* from 16 to 256 does not significantly impact the effectiveness of the local outlier detection methods.
> > >
> > > |   Trigger   | Poisoning rate |  ABL  |   CD  | SafeCLIP |  LID  | iForest |  DAO k=16 | DAO k=32 | DAO k=64 | DAO k=128 | DAO k=256 |
> > > |:-----------:|:--------------:|:-----:|:-----:|:--------:|:-----:|:-------:|:---------:|:--------:|:--------:|:---------:|:---------:|
> > > |    Patch    |      0.01%     | 27.86 | 97.17 |   83.42  | 99.29 | `99.73` | **99.86** |   99.78  |   99.75  |   99.71   |  `99.70`  |
> > > | Clean Label |      0.07%     | 63.50 | 48.68 |   46.93  | 88.23 | `94.01` | **97.06** |   96.81  |   96.30  |   95.83   |  `95.38`  |
> > > |  Nashville  |      0.1%      | 61.69 | 98.33 |   46.37  | 61.07 | `99.35` | **99.62** |   99.59  |   99.54  |   99.49   |  `99.44`  |
> > > |    WaNet    |      0.1%      | 56.07 | 99.19 |   85.82  | 57.07 | `99.55` | **99.85** |   99.84  |   99.82  |   99.80   |  `99.77`  |
> > > |    Blend    |      0.1%      | 60.07 | 99.64 |   57.01  | 54.94 | `99.80` | **99.88** |   99.86  |   99.84  |   99.82   |  `99.80`  |
> > > |     SIG     |      0.1%      | 56.88 | 99.06 |   82.15  | 54.03 | `99.62` | **99.69** |   99.68  |   99.67  |   99.66   |  `99.64`  |
> > > |     MT-S    |      0.1%      | 47.32 | 99.34 |   80.98  | 94.87 | `99.59` | **99.77** |   99.75  |   99.71  |   99.68   |  `99.63`  |
> > > |     MT-M    |      0.1%      | 53.24 | 95.33 |   78.10  | 97.11 | `98.32` | **99.04** |   99.00  |   98.90  |   98.77   |  `98.66`  |
> > >
> > > ---
> > >
> > > **Q2:** Further explanation of density.
> > >
> > > **A2:**
> > >
> > > > From my understanding, this is not related to label information. Once poisoned, the feature distribution is there.
> > >
> > > Yes, our detection does not rely on any label information. We point this out to distinguish our work from the related work mentioned by the reviewer.
> > >
> > > > These 100 examples will cluster together densely due to the common trigger feature. In this setting, do you mean the poisoned examples have higher density but lower density ratio? If it is, how to explain this?
> > >
> > > In the mentioned example, the 100 backdoored samples are tightly clustered within the representation space. The observed density, density ratio, and simplified local outlier factors (SLOF) depend on the size of the local neighborhood $k$. Specifically:
> > >
> > > - When $k$ is less than 100, the SLOF score of backdoor samples is similar to that of clean samples.
> > > - When $k$ exceeds 100, the SLOF score of backdoor samples becomes significantly higher than that of clean samples, as clean samples fall within the local neighborhood of the backdoor samples.
> > >
> > > This behavior explains the observed $\frac{k}{b}$ ratio mentioned in our initial response to Q1/A1.
> > >
> > > To aid our discussion, we have updated the draft to include a visualization of the embedding space using 30 backdoor samples (visualizing 100 backdoor samples proved too complex). The poisoning ratio in this visualization is 2.9%. We invite the reviewer to refer to Figure 8 in Appendix B.5 for illustration. Note in this figure that the density ratio is represented as the radius to the solid line over dash lines for both clean(green) and backdoor (red) samples. The SLOF score is the average density ratio within the $k$  local neighborhood.
> > >
> > > This visualization is based on the same randomly sampled batch, with backdoor samples deliberately added as part of a controlled experiment to support the discussion above.

---

> > > > ### Author Response · Authors · 2024-11-23
> > > > **Response to Reviewer 5XDB**
> > > >
> > > > **Q3:** Results on the adaptive attack.
> > > >
> > > > **A3:**
> > > >
> > > > > This is counterintuitive. How to explain this?
> > > >
> > > > We acknowledge that this result may seem surprising; however, it was indeed what we observed.
> > > >
> > > > We believe this phenomenon arises because, during pre-training with an adaptive objective, the SLOF scores for backdoor samples are minimized only within the specific neighborhood of samples in the mini-batch. This minimization does not generalize to the entire embedding space. As a result, when detection is performed using randomly sampled data points from the broader pool to identify potential neighbors, the density profile of backdoor samples fails to align with that of clean samples.
> > > >
> > > > For an adaptive strategy to fully evade detection, it would need to account for all possible combinations of local neighbors, which is computationally infeasible. Furthermore, such an approach could conflict with the primary CLIP objective, undermining the model's performance.
> > > >
> > > > Finally, we emphasize that the adaptive attack experiment is not realistic for attackers aiming to evade detection. In practical data poisoning scenarios, attackers are generally limited to poisoning the training data and cannot directly influence the training process in an adaptive manner. Achieving precise manipulation of density in the embedding space using poisoning alone is even more challenging.

---

### Official Review · Reviewer_p5qZ · 2024-11-03

**Soundness:** 3
**Presentation:** 3
**Contribution:** 2
**Rating:** 6
**Confidence:** 2

**Summary:**

This study exploits this feature to detect backdoor samples by analyzing the local spatial representation features of backdoor samples learned by the CLIP model and finding that the local neighborhoods of these samples are sparser than clean samples. Specifically, the paper proposes the use of traditional density-ratio based local anomaly detection methods, such as Simplified Local Anomaly Factor (SLOF) and Dimension-Aware Anomaly Detection (DAO), to efficiently detect CLIP backdoor samples. These methods are able to detect backdoor samples from large-scale network datasets and clean up the dataset.

**Strengths:**

EFFICIENCY: The method can quickly detect backdoor samples in large-scale datasets. Using 4 Nvidia A100 GPUs, a network dataset of millions (e.g., CC3M) can be cleaned in 15 minutes, which is especially important for processing large-scale datasets.

Accuracy: The proposed methods, especially the density-based local anomaly detection methods (e.g., SLOF and DAO), show high accuracy in detecting CLIP backdoor samples. These methods are able to effectively distinguish backdoor samples from normal samples, even at very low cast rates (e.g., 0.01%).

Robustness: the method shows good stability and robustness at different poisoning rates, especially at low poisoning rates. Even at poisoning rates as high as 10%, the detection performance can still be maintained at a high level by adjusting the localization parameter k.

**Weaknesses:**

Sensitivity to parameters: local anomaly detection methods (e.g., SLOF and DAO) rely on the choice of the localizability parameter k. Although the paper mentions that these methods are relatively robust to the value of k, improper parameter selection may still affect detection performance.

Dataset dependency: the method performs well on the CC3M dataset, but its validity on other datasets may require further validation, as different datasets may have different characteristics and distributions.

Experiments may be based primarily on specific model architectures (e.g., ResNet-50 and ViT-B-16). Models with different architectures may have different sensitivities to backdoor attacks, so testing on multiple model architectures may provide a more comprehensive evaluation of the approach.

**Questions:**

Does this new backdoor defense approach apply to other datasets like Wikipedia-based Image Text (WIT) and RedCaps?

The paper presents advantages over traditional anomaly detection methods but may lack a comparison with the latest or state-of-the-art backdoor detection methods. This may limit the full understanding of method performance.

---

> ### Author Response · Authors · 2024-11-19
> **Response to Reviewer p5qZ (1/2)**
>
> Thank you for taking the time to review our paper and for your valuable comments. Please find our responses to your questions below.
>
> ---
>
> **Q1:** Sensitivity to parameters
>
> **A1:** Thank you for raising the concern. We have found that the choice of $\( k \)$ has minimal impact on the detection performance. For easier comparison, we have combined the results from Table 1 and the ablation study on $\( k \)$ in Appendix B.2 into a single table below. It shows that even when using the $\( k \)$ value that yields the lowest detection performance (DAO with $\( k = 256 \)$), our method still surpasses the best baseline method (iForest) in most cases. The only exception is with the patch trigger, where iForest slightly exceeds DAO by a margin of only 0.03%. This demonstrates that our method is robust to the choice of $\( k \)$ and maintains superior performance across a range of parameter values.
>
> | Trigger | ABL | CD | SafeCLIP | LID | iForest | DAO k=16 | DAO k=32 | DAO k=64 | DAO k=128 | DAO k=256 |
> |:---:|:---:|:---:|:---:|:---:|:---:|:---:|:---:|:---:|:---:|:---:|
> | Patch | 27.86 | 97.17 | 83.42 | 99.29 | `99.73` | **99.86** | 99.78 | 99.75 | 99.71 | `99.70` |
> | Clean Label | 63.50 | 48.68 | 46.93 | 88.23 | `94.01` | **97.06** | 96.81 | 96.30 | 95.83 | `95.38` |
> | Nashville | 61.69 | 98.33 | 46.37 | 61.07 | `99.35` | **99.62** | 99.59 | 99.54 | 99.49 | `99.44` |
> | WaNet | 56.07 | 99.19 | 85.82 | 57.07 | `99.55` | **99.85** | 99.84 | 99.82 | 99.80 | `99.77` |
> | Blend | 60.07 | 99.64 | 57.01 | 54.94 | `99.80` | **99.88** | 99.86 | 99.84 | 99.82 | `99.80` |
> | SIG | 56.88 | 99.06 | 82.15 | 54.03 | `99.62` | **99.69** | 99.68 | 99.67 | 99.66 | `99.64` |
> | MT-S | 47.32 | 99.34 | 80.98 | 94.87 | `99.59` | **99.77** | 99.75 | 99.71 | 99.68 | `99.63` |
> | MT-M | 53.24 | 95.33 | 78.10 | 97.11 | `98.32` | **99.04** | 99.00 | 98.90 | 98.77 | `98.66` |
>
>
> ---
>
> **A2:** Dependency on the dataset.
>
> **Q2:**
> Thanks for your thoughtful comment. Please refer to Appendix B.7 Table 10, where we already conducted our experiments on CC12M, which shows a consistent level of detection performance.
>
>
> ---
>
> **Q3:** Based primarily on specific model architectures and comprehensive evaluation of different architectures.
>
> **A3:**
> Thanks for your valuable comment. We conducted an additional experiment using ResNet-101 as the encoder for detection, with all other experimental settings consistent with the initial submission. The results indicate that the choice of architecture does not impact detection performance.
>
> In addition to the commonly used ResNet and ViT, please let us know if there is a specific architecture of interest that you believe would further validate our findings. We would be happy to include it in our discussion during the rebuttal period. However, due to time constraints, we are currently unable to test all possible variants of ResNet and ViT. We plan to extend our analysis to include a wider range of architectures in the next version of our paper to provide a more comprehensive evaluation.
>
> |  Encoder  | Poisoning Rate | Clean Acc |  ASR  |  ABL  |   CD  | SafeCLIP |  LID  | iForest | $k$-dist |    SLOF   |    DAO    |
> |:---------:|:--------------:|:---------:|:-----:|:-----:|:-----:|:--------:|:-----:|:-------:|:--------:|:---------:|:---------:|
> |  ResNet50 |      0.01%     |    17.0   | 100.0 | 27.86 | 97.17 |   83.42  | 99.29 |  99.73  |   99.75  | **99.86** | **99.86** |
> | ResNet101 |      0.01%     |    17.6   | 100.0 | 26.69 | 93.85 |   82.22  | 99.28 |  99.54  |   99.53  | **99.62** | **99.62** |
>
> ---
>
> **Q4:** Applications on other datasets.
>
> **A4:** Thanks for your suggestion. Our proposed method is designed to work with any image-text pair dataset. As mentioned in our response to Q2/A2, we have demonstrated that our detection method is generalizable on the CC12M dataset.
>
> Unfortunately, due to time constraints, we were unable to download and conduct experiments on the RedCaps and WITdatasets, which contain 12 million and 38 million image-text pairs, respectively. We only managed to download 30% of the RedCaps as of today. We will do our best to provide a comparison before the rebuttal ends.

---

> ### Author Response · Authors · 2024-11-19
> **Response to Reviewer p5qZ (2/2)**
>
> **Q5:** lack of comparison with the latest or state-of-the-art backdoor detection methods
>
> **A5:**
>
> > may lack a comparison with the latest defense
>
> To the best of our knowledge, SafeCLIP (Yang et al., 2024) is the only backdoor data detection method specifically designed for CLIP backdoor poisoning attacks, and it represents the latest advancement in this area.
>
> > may lack a comparison with the state-of-the-art backdoor detection methods
>
>
> We also included a comparison with Cognitive Distillation (CD) (Huang et al., 2023), a state-of-the-art backdoor detection method originally designed for supervised learning. CD can be adapted for CLIP backdoor detection, whereas other supervised learning methods require class labels or classification predictions, which makes them unsuitable for our context.
> Comparisons with both SafeCLIP and CD are presented in Table 1 of our initial submission, where local outlier detection methods demonstrate clear advantages in effectiveness.
>
>  If there are other backdoor detection methods for CLIP that we may have overlooked, we welcome reviewer recommendations and are happy to include additional comparisons in future work. Thank you for raising this important point.
>
> ---
>
> [1] Yang, Wenhan, Jingdong Gao, and Baharan Mirzasoleiman. "Better Safe than Sorry: Pre-training CLIP against Targeted Data Poisoning and Backdoor Attacks." In ICML 2024.
>
> [2] Huang, Hanxun, et al. "Distilling Cognitive Backdoor Patterns within an Image." In ICLR 2023.

---

> ### Comment · Reviewer_p5qZ · 2024-11-23
>
> Thanks for the effort in downloading the RedCaps dataset and addressing comparisons with SafeCLIP and Cognitive Distillation. However, there are additional related works that are highly relevant to backdoor detection in pre-trained encoders and multimodal models like CLIP.
>
> Incorporating these works into your discussion would provide a more comprehensive comparison and situate your contribution within the broader context of recent advancements in this field, including DECREE, BDetCLIP, and Adversarial Backdoor Defense (see references below). A broader evaluation of these approaches, particularly in terms of their effectiveness, computational efficiency, and applicability, would strengthen this work.
>
> [1] Kuang, Junhao, et al. "Adversarial backdoor defense in clip." arXiv preprint arXiv:2409.15968 (2024)
>
> [2] Niu, Yuwei, et al. "Bdetclip: Multimodal prompting contrastive test-time backdoor detection." arXiv preprint arXiv:2405.15269 (2024).
>
> [3] Feng, Shiwei, et al. "Detecting backdoors in pre-trained encoders." Proceedings of the IEEE/CVF Conference on Computer Vision and Pattern Recognition. 2023.

---

> > ### Author Response · Authors · 2024-12-03
> > **Thanks for your feedback.**
> >
> > Thanks for your feedback. We have already included these papers in the discussion of related works in our updated draft.
> >
> > - DECREE and Adversarial Backdoor Defense focus on backdoor model detection, whereas our work targets backdoor data detection, making direct comparisons infeasible.
> > - BDetCLIP requires class labels and performs data detection at inference time, which differs from our pre-training stage detection and does not rely on the class label. This complicates direct comparisons.
> >
> > Additionally, we would like to note that Adversarial Backdoor Defense and BDetCLIP are contemporaneous works.
> >
> > ---
> >
> > We have downloaded the RedCaps and evaluated the detection using this dataset. The conclusion is the same as using CC3M and CC12M. Please find the results on the Patch trigger below. We will extend the results to all triggers in the next revision.
> >
> > | Clean Acc | ASR | ABL | CD | SafeCLIP | LID | iForest | $k$-dist | SLOF | DAO |
> > |:---:|:---:|:---:|:---:|:---:|:---:|:---:|:---:|:---:|:---:|
> > | 29.94 | 96.90 | 65.73 | 93.38 | 74.27 | 69.48 | 95.73 | 95.76 | 95.80 | **95.81** |

---

### Official Review · Reviewer_6Fx7 · 2024-11-03

**Soundness:** 2
**Presentation:** 2
**Contribution:** 2
**Rating:** 6
**Confidence:** 4

**Summary:**

Summary: In this paper, the authors analyzed the representations of backdoor-poisoned samples learned by CLIP models and aim to design an efficient detection method for those backdoors. The experiments reveal that an unintentional backdoor already exists in the original CC3M dataset and has been trained into a popular open-source model released by OpenCLIP.

**Strengths:**

+ Proposed a backdoor data detection method for CLIP models
+ Experimental results show a high detection rate on the tested models.

**Weaknesses:**

- The technical novelty is very limited as the main idea of this paper is simply applying existing outlier detection metrics on detecting CLIP backdoor samples. I didn’t see a major technical innovation in this process.

- They only considered the simple and naive attack settings (directly poisoning the data with fixed noise), which seems reasonable to use those common outlier metrics to detect the possible poisoned data. It is not clear whether optimized trigger can be detected, for example  [1]. The authors might want to compare and comment on those more complicated attacker settings.

[1] Sun, Weiyu, et al. "Backdoor Contrastive Learning via Bi-level Trigger Optimization." ICLR2024


- It is also concerning especially if we are considering the adaptive attack setting where the attacker knows your defense strategy. It seems not hard to circumvent the current metric and check the detector by having an additional constraint forcing the density/metrics to look normal.

- The authors claim their method to be efficient, yet I didn’t find detailed comparison/ experimental results suggesting or comparing the runtime with other baselines. Since quite a few detection metrics rely on the k-nearest neighbor, I am confused about how it could be an efficient strategy when the number of data samples is large.

**Questions:**

see above

---

> ### Author Response · Authors · 2024-11-19
> **Response to Reviewer 6Fx7 (1/2)**
>
> Thank you very much for reviewing our paper and the valuable comments. Please find our response to your questions below:
>
> ---
> **Q1:** Technical novelty
>
> **A1:**
> According to the ICLR Reviewer Guide (2021–2025), novelty should be evaluated not only based on technical methods but also on novel findings. We believe our work offers significant contributions in both respects.
>
> We summarize our novelty and contributions as follows:
>
> 1. **Discovery of Local Outliers in Deep Representation Space**: We found that backdoor data targeting CLIP models manifest as local outliers in the deep representation space. This insight led us to develop a detection method using local outlier analysis.
>
> 2. **Revelation of Real-World Backdoor Data**: We uncovered an actual backdoor trigger and a collection of backdoor data sourced from the Internet, which are already being incorporated into pretraining datasets.
>
> **These novel findings are important to the ICLR community.**
>
> We would like to direct the reviewer to the ICLR 2023 workshop on “Backdoor Attacks and Defenses in Machine Learning,” specifically the invited talk “Why Nobody is Using Your Backdoor Defense” by Vitaly Shmatikov [1]. The talk highlighted two key issues:
>
> 1. The lack of real-world examples of backdoors.
> 2. The practical complexity of existing defense methods.
>
> Our work addresses these concerns by demonstrating that:
>
> 1. **Backdoor Data Are Present in Real-World Datasets**: We provide evidence that backdoor data are already present on the Web and are being used in pretraining, thus filling the gap of real-world examples.
>
> 2. **Simple and Effective Defense Solutions**: We propose a straightforward yet effective method for mitigating backdoor threats, countering the complexity of existing defense approaches.
>
> In practical settings, data cleansing is an essential part of machine learning pipelines (e.g., Amazon SageMaker). Our method offers an additional step to enhance this process by efficiently detecting and filtering backdoor data, thereby mitigating the real-world backdoor threat.
>
> To the best of our knowledge, no current backdoor data filtering methods can efficiently clean million-scale pretraining datasets. We genuinely hope the community can embrace simple yet effective solutions to address this realistic safety threat to existing pretraining paradigms.
>
> [1] Invited Talk by Vitaly Shmatikov at the ICLR 2023 workshop: [https://iclr.cc/virtual/2023/workshop/12825](https://iclr.cc/virtual/2023/workshop/12825)
>
> ---
>
> **Q2:** Lack of comparison with optimized trigger BLTO
>
> **A2:**
> Thanks for your suggestion. We have conducted additional experiments using the checkpoints provided in the BLTO open-source repository, specifically those trained on CIFAR-10 and ImageNet-100. All other experimental settings remained consistent with those in our initial submission.
>
> The detection results, presented in the table below, demonstrate that BLTO triggers can also be effectively detected by our method. This confirms that our approach is robust against optimized triggers like BLTO. Therefore, all the conclusions drawn in our initial submission apply to BLTO as well.
>
> |      Trigger     | Poisoning Rate | Clean Acc |  ASR  |  ABL  |   CD  | SafeCLIP |  LID  | iForest | $k$-dist |    SLOF   |    DAO    |
> |:----------------:|:--------------:|:---------:|:-----:|:-----:|:-----:|:--------:|:-----:|:-------:|:--------:|:---------:|:---------:|
> |   BLTO_CIFAR10   |      0.1%      |   16.72   | 98.34 | 58.88 | 97.60 |   84.25  | 43.04 |  99.81  |   99.85  | **99.86** | **99.86** |
> | BLTO_ImageNet100 |      0.1%      |   16.79   | 96.96 | 55.21 | 98.76 |   84.66  | 45.52 |  99.74  |   99.77  |   99.77   | **99.78** |

---

> ### Author Response · Authors · 2024-11-19
> **Response to Reviewer 6Fx7 (2/2)**
>
> **Q3:** Lack of analysis on adaptive attack
>
> **A3:**
>
>  Thank you for your thoughtful suggestion. We have conducted additional experiments to analyze adaptive attacks that aim to circumvent our detection method. Specifically, we tested an adaptive attack using the following optimization objective during pretraining:
>
> $$
> \mathcal{L}_{\text{CLIP}}(\mathbf{z}^x, \mathbf{z}^t) + \text{SLOF}(\mathbf{z}^{x'}), \quad \mathbf{x}' \in \mathcal{D}_b,
> $$
>
> where $\mathbf{z}^x$ and $\mathbf{z}^t$ are the image and text embeddings, respectively, and $\text{SLOF}(\mathbf{z}^{x'})$ denotes the Local Outlier Factor score of the backdoor-poisoned samples $\mathbf{x}'$ in the backdoor dataset $\mathcal{D}_b$. This objective allows the attacker to minimize the outlier score of the poisoned samples during pretraining. We implemented this adaptive attack using the **BadNets** patch trigger, keeping all other experimental settings consistent with those in our initial submission.
>
> The results are reported in the table below. It clearly shows that this adaptive strategy does not circumvent our detection method; in fact, it even improves our detection performance. This is because forcing the poisoned backdoor samples to mimic the density profile of clean samples is only effective within a specific neighborhood in the feature space. To fully evade detection, an attacker would need to account for all possible neighborhoods generated by various combinations of data points—a task that is computationally infeasible given the scale of web datasets, which often contain millions or even billions of samples. Therefore, our detection method remains robust even against adaptive attacks that attempt to minimize the outlier scores of poisoned samples. This reinforces the effectiveness of our approach in real-world settings where attackers may employ sophisticated strategies to hide backdoor triggers.
>
> |      Method     | Trigger | Poisoning Rate | Clean Acc |  ASR  |  $k$-dist |    SLOF   |    DAO    |
> |:---------------:|:-------:|:--------------:|:---------:|:-----:|:---------:|:---------:|:---------:|
> |     Standard    |  Patch  |      0.01%     |   17.00   | 100.0 |   99.75   | **99.86** | **99.86** |
> | Adaptive Attack |  Patch  |      0.01%     |   15.94   | 100.0 | **100.0** | **100.0** | **100.0** |
>
> ---
>
> ---
>
> **Q4:** The time efficiency comparison with baseline and scaling with number of samples.
>
> **A4:**
> Thank you for your question regarding the time efficiency of our method. Please refer to Appendix B.1, Table 3 in our initial submission for a detailed time efficiency comparison. For your convenience, we have copy-pasted the results to the table below.
>
> Table: Time efficiency comparison of different methods (measured in number of hours).
>
> | ABL |  CD  | SafeCLIP | LID | iForest | $k$-dist | SLOF | DAO |
> |:---:|:----:|:--------:|:---:|:-------:|:--------:|:----:|:---:|
> | 4.1 | 11.2 |    0.2   | 0.2 |   0.3   |    0.2   |  0.2 | 0.2 |
>
> > metrics rely on the k-nearest neighbor
>
> For each data point, we sample a batch of data as the pool to select the nearest neighbors, as explained in Appendix A. Calculating the k-nearest neighbors involves a single matrix multiplication to compute pairwise distances, followed by sorting these distances. This operation is highly efficient on GPUs, taking approximately 0.6 seconds to process a batch of 2,048 data points, including the time for extracting embeddings using the CLIP encoder.
>
> >  I am confused about how it could be an efficient strategy when the number of data samples is large.
>
> Our approach scales linearly with the number of data points $\( n \)$, having a time complexity of $\( \mathcal{O}(n) \)$. In comparison:
>
> - **ABL** (Automatic Backdoor Detection) has a time complexity of $\( \mathcal{O}(mn) \)$, where $\( m \)$ is the number of epochs needed to track sample-specific losses.
> - **CD** (Cognitive Distillation) requires $\( \mathcal{O}(tn) \)$ time, where $\( t \)$ represents the number of optimization steps per sample.
> - **Isolation Forest** (iForest) requires $\( \mathcal{O}(hn) \)$ time, where $\( h \)$ represents the number of trees in the ensemble.
>
> While methods like **SafeCLIP** also have linear time complexity $(\( \mathcal{O}(n) \))$, they are less effective than our method in detecting backdoor data.
>
> Therefore, our method offers a favorable balance between efficiency and effectiveness, even when dealing with large-scale datasets. The linear scalability ensures that our approach remains practical and efficient as the number of samples increases, addressing concerns about processing time in large datasets.

---

> > ### Comment · Reviewer_6Fx7 · 2024-11-19
> > **Thanks for the response**
> >
> > I thank the authors for their detailed response. The additional experiments on BLTO and adaptive attacks addressed my corresponding concerns and I would like to raise my score.
> >
> > For time complexity, I feel it is a bit misleading to call in linear in data points n since the linear time calculation didn't count the K-nearest neighbor matrix calculation (if I understand correctly, correct me if not). Despite that the authors claim it is efficient on GPUs, it is not neglectable especially when n is large. I think that part should also be counted towards the time complexity calculation and should be reported clearly. Again, since the authors claim that it is for large pre-training datasets, it could have a huge influence if we are talking about billions of data points (way larger than the current experiment size).

---

> > > ### Author Response · Authors · 2024-11-20
> > > **Further clarification**
> > >
> > > We greatly appreciate your prompt response! We understand your concern about the time complexity. Please allow us to provide further clarification.
> > >
> > > ---
> > >
> > > If we consider the whole dataset contains $n$ number of data points, for each data point, our detection looks at a batch of the data points sampled from the dataset as potential neighbors, with batch size $b$. The outlier score is calculated with respect to all other points within the batch, not with respect to the entire dataset.
> > >
> > > > Linear time calculation didn't count the K-nearest neighbor matrix calculation
> > >
> > > The K-nearest neighbor matrix calculation is with respect to a batch of data with size $b$ rather than the entire dataset $n$.
> > >
> > > > Despite that, the authors claim it is efficient on GPUs, it is not neglectable, especially when n is large. I think that part should also be counted towards the time complexity calculation and should be reported clearly.
> > >
> > > Our claim in the initial response that it is efficient on GPUs is based on the batch of $b$ data points. It takes approximately 0.6 seconds to process a batch of 2,048 data points, including the time of extracting the embedding with the encoder. Given that the entire batch of data points is randomly sampled, and the K-nearest neighbor matrix contains pairwise distance, the outlier scores for all 2,048 data points can be computed together within 0.6 seconds. This time cost is constant regardless of the dataset size $n$.
> > >
> > > Considering the large-scale pre-training dataset with size $n$, **the time needed for processing a batch of $b$ data points is constant with respect to $n$**, so to process the entire dataset, we just need to perform an iteration over the entire dataset in batches. Hence, the time complexity is linearly scaled with $n$.
> > >
> > > >  Again, since the authors claim that it is for large pre-training datasets, it could have a huge influence if we are talking about billions of data points (way larger than the current experiment size).
> > >
> > > We appreciate your insightful observation of the potential influence on large-scale datasets. In our submission, we have experimented with 3 million and 12 million datasets. The time cost of running outlier scores calculation is 15 minutes and 60 minutes on 4 GPUs, respectively. This further indicates the linear scaling with respect to the number of data points. For 1 billion data points, it would require approximately 83 hours. It is also worth noting that the outlier scores can be calculated in parallel; with more GPUs, the wall time can be further reduced. We believe this is very efficient and practical for large-scale pre-training datasets.

---

> > > > ### Comment · Reviewer_6Fx7 · 2024-11-22
> > > > **Thanks**
> > > >
> > > > Thanks for the further explanation and I think it addresses my concern. I would raise my score to 6

---

### Author Response · Authors · 2024-11-19
**Summary of changes in the revision**

We have made the following updates to the draft based on the reviewers’ valuable suggestions:

- As suggested by Reviewer **6Fx7**, we included the BLTO trigger in our evaluation in Table 1. The results demonstrate that our detection method accurately identifies data poisoned by the BLTO trigger.
- We added two references to related works: one on the BLTO trigger and another on the density study of backdoor data, as mentioned by Reviewer **5XDB**.
- To address concerns raised by Reviewers **6Fx7** and **5XDB**, we included a detailed analysis of white-box adaptive attacks in Appendix B.6. The results show that even with white-box access to the pretraining process, adaptive attacks are insufficient to evade our detection method.

---

### Meta-Review · Area_Chair_YSA4 · 2024-12-13

**Metareview:**

Scientific Claims and Findings:
The paper identifies that backdoor samples in CLIP models exhibit unique local subspace sparsity, enabling efficient backdoor detection. It demonstrates the method’s effectiveness by detecting backdoors in CC3M for OpenCLIP in 15 minutes on 4 A100 GPUs, outperforming existing methods.

Strengths:
Simple and effective method with strong empirical results on a large dataset (CC3M).
Computationally efficient, suitable for practical use.

Weaknesses:
The adaptive attacks studied in the paper/rebuttal appear to increase detection rates rather than bypassing the method, suggesting a weak adaptive attack (Section B.6). The meta reviewer suggest the authors to also try other adaptive attacks in the revision, say generating backdoors samples so that they locally become dense, bypassing the detection. A further discussion and experiments are suggested in the revision.

Recommendation:
Acceptance based on the strengths that are acknowledged by all the reviewers. Adaptive attacks could be improved. The method is still worth publishing despite the weakness on the adaptive attacks.

**Additional Comments On Reviewer Discussion:**

Reviewers find the author addressed or partially addressed their points, with final all positive reviews.

All reviewers ask adaptive attacks, which worthy further discussion and experiments given that the adaptive attack evaluate does not seem to be strong to bypass the defense better than vanilla attacks.

---

### Decision · Program_Chairs · 2025-01-22

Accept (Poster)